# Neutrophil microvesicles drive atherosclerosis by delivering *miR-155* to atheroprone endothelium

Ingrid Gomez[1,11], Ben Ward[1,2,11], Celine Souilhol[1,2,11], Chiara Recarti[1,3], Mark Ariaans [1], Jessica Johnston [1], Amanda Burnett[1], Marwa Mahmoud[1,4], Le Anh Luong[5], Laura West [1], Merete Long[1], Sion Parry[6], Rachel Woods [6], Carl Hulston [6], Birke Benedikter [7], Chiara Niespolo[1], Rohit Bazaz[1], Sheila Francis [1], Endre Kiss-Toth [1], Marc van Zandvoort[3], Andreas Schober [8], Paul Hellewell [1,9], Paul C. Evans[1,2,10] & Victoria Ridger [1,2]*

Neutrophils are implicated in the pathogenesis of atherosclerosis but are seldom detected in atherosclerotic plaques. We investigated whether neutrophil-derived microvesicles may influence arterial pathophysiology. Here we report that levels of circulating neutrophil microvesicles are enhanced by exposure to a high fat diet, a known risk factor for athero-sclerosis. Neutrophil microvesicles accumulate at disease-prone regions of arteries exposed to disturbed flow patterns, and promote vascular inflammation and atherosclerosis in a murine model. Using cultured endothelial cells exposed to disturbed flow, we demonstrate that neutrophil microvesicles promote inflammatory gene expression by delivering *miR-155,* enhancing NF-κB activation. Similarly, neutrophil microvesicles increase *miR-155* and enhance NF-κB at disease-prone sites of disturbed flow in vivo. Enhancement of atherosclerotic plaque formation and increase in macrophage content by neutrophil microvesicles is dependent on *miR-155.* We conclude that neutrophils contribute to vascular inflammation and atherogenesis through delivery of microvesicles carrying *miR-155* to disease-prone regions.

[1] Department of Infection, Immunity and Cardiovascular Disease, University of Sheffield, Sheffield, UK. [2] INSIGNEO Institute for In Silico Medicine, University of Sheffield, Sheffield, UK. [3] Department of Molecular Cell Biology, CARIM School for Cardiovascular Diseases, Maastricht University, Maastricht, The Netherlands. [4] Cardiovascular Mechanobiology and Nanomedicine, Department of Medicine, Emory University, Atlanta, GA, USA. [5] William Harvey Research Institute, Queen Mary University, London, UK. [6] School of Sport, Exercise and Health Sciences, Loughborough University, Loughborough, UK. [7] Department of Medical Microbiology, NUTRIM School of Nutrition and Translational Research in Metabolism, Maastricht University, Maastricht, The Netherlands. [8] Experimental Vascular Medicine, Ludwig-Maximilian University of Munich, Munich, Germany. [9] College of Health and Life Sciences, Brunel University, London, UK. [10] Bateson Institute, University of Sheffield, Sheffield, UK. [11] These authors contributed equally: Ingrid Gomez, Ben Ward, Celine Souilhol. *email: V.C.Ridger@Sheffield.ac.uk

A causal role for neutrophils in atherosclerosis is now evident and these abundant leucocytes have been shown to play a part both in plaque development[1,2] and plaque erosion[3], as well as being implicated in plaque rupture[4,5]. Increased levels of circulating neutrophils exacerbate atherosclerotic plaque formation in mice[2] and indirect evidence also links increased circulating leucocyte counts and infection with an increased risk of cardiovascular disease[6,7]. Nevertheless, detection of neutrophils in lesions is rare, possibly due to their short life span, the rapid removal of senescent neutrophils by macrophages within the developing plaque, and/or the lack of a highly specific detection method (see ref. [8] for review). In order to address this paradox, we have investigated whether neutrophils exacerbate vascular inflammation through the release of pro-inflammatory microvesicles (MVs), thus influencing atherosclerotic plaque formation without entering the vessel wall.

MVs are 0.1–1 µm vesicles released from the cell membrane in response to various stimuli or during apoptosis. Depending on the cell source, MVs vary in their size, content, and surface marker expression[9]. While MVs are present in healthy individuals[10], increased levels of leucocyte MVs have been observed in patients with sepsis[11], acute vasculitis[12] and individuals with high risk of cardiovascular disease[13]. Advances in our understanding of extracellular vesicle function have led to the discovery of a novel mechanism by which cells can communicate with each other through the transfer of vesicle cargo, such as noncoding RNA (e.g. microRNA (miRNAs)), to target cells[14–17]. Neutrophil-derived microvesicles (NMVs) have been detected in human atherosclerotic plaques[18] but their role in plaque progression has not been studied.

Atherosclerotic plaque distribution is focal, with plaques developing at sites of disturbed flow, such as bifurcations, where adhesion molecules are highly expressed[19,20]. Disturbed flow generates shear stress (mechanical drag) that is low and oscillatory at atheroprone sites, whereas shear stress is high at atheroprotected sites[21,22]. Vascular inflammation is regulated by a number of transcription factors including the master regulator nuclear factor (NF)-κB[23], the expression of which is known to be enhanced in atheroprone regions[24–26] and induced by disturbed flow[27]. Enhanced vascular inflammation leads to leucocyte recruitment to these sites, with the presence of monocytes within the vessel wall a characteristic of atherosclerotic plaque development.

Here we show that NMVs are released in response to high-fat diet and preferentially adhere to sites prone to atherosclerotic plaque development. We also demonstrate NMVs contain miRNAs and are internalised by arterial endothelial cells. NMVs induce NF-κB expression, through delivery of cargo such as miRNAs, leading to enhanced endothelial inflammation, monocyte recruitment and atherosclerotic plaque development.

## Results

### Characterisation of isolated NMVs.
In order to characterise the subpopulation of extracellular vesicles investigated in these studies, we performed transmission electron microscopy of MVs derived from both human (Fig. 1a, b) and mouse (Fig. 1d, e) peripheral blood neutrophils. This demonstrated the heterogeneity and structure of isolated NMVs. Tunable Resistive Pulse Sensing (TRPS) was performed on human NMVs (Fig. 1c; mode size 280 ± 16.6 nm (SEM)) and Nanoparticle Tracking Analysis (NTA) performed on mouse NMVs (Fig. 1f; mode size 165 ± 7.5 nm (SEM)) to assess the size distribution.

### Proatherogenic diet elevates NMV levels.
We determined whether exposure to a high-fat diet in healthy human subjects affected circulating levels of NMVs. The energy intake and diet composition is described in the methods and an example of the typical daily food intake is shown in Supplementary Table 1. Flow cytometry analysis revealed that human plasma NMV levels were significantly increased after 1 week of high-fat feeding (~27% increase, Fig. 1g) indicating that a high-fat diet induced increased circulating NMV levels. Analysis of markers of different cellular origins revealed that MVs derived from neutrophils, platelets and monocyte, but not endothelial cells, were significantly increased after high-fat feeding (Supplementary Tables 2 and 3 and Supplementary Fig. 1a). However, the overall distribution of MVs from different cell types was not altered (Supplementary Fig. 1b). We also found elevated levels of total plasma MVs in $ApoE^{-/-}$ mice on high-fat diet compared to chow (Fig. 1h), however due to technical difficulties with antibody labelling we were unable to differentially label NMVs directly in the plasma of mice. We therefore determined the effect of depleting neutrophils from the circulation and found a significant reduction in circulating MV levels compared to control (~32%, Fig. 1i). Taken together, these findings provide evidence that NMV are produced in vivo in response to a proatherogenic diet.

### NMVs preferentially adhere to atheroprone regions.
Having determined that high-fat diet induced production of NMVs, we investigated whether these endogenously released NMVs were detectable in the vessel wall. Flow cytometry analysis of aortic arch homogenates from $ApoE^{-/-}$ mice fed chow or a Western diet revealed that greater numbers of NMVs were detected in the vessel wall at 20 weeks compared to 6 weeks (Fig. 1j), suggesting that NMVs accumulate at atheroprone regions. Significantly more platelet and monocyte but not endothelial cell derived MVs were also detected in the homogenates but, similar to the human responses to high-fat feeding, the overall distribution of MVs from different cell types was not altered (Supplementary Table 4 and Supplementary Fig. 2) at 20 weeks. In order to investigate the mechanisms by which NMVs are recruited to the vessel wall, we determined whether NMVs were able to adhere to arteries in vivo. Fluorescently labelled NMVs ($4 \times 10^6$) or supernatant from fluorescently labelled NMV pellets was injected via the tail vein into $ApoE^{-/-}$ mice that had been fed a Western diet for 6 weeks. This number of NMVs is similar to the 30% increase in circulating NMVs observed in human subjects after 7 days on an atherogenic diet (Fig. 1g). Using en face confocal microscopy of the inner and outer curvature of the aorta of each injected mouse, fluorescently labelled NMVs were rarely detected in atheroprotected regions (outer curvature of aortic arch; Fig. 2a, b) after 2 h but significantly higher numbers were detected at the atheroprone regions (inner curvature of aortic arch; Fig. 2c, d; quantified in Fig. 2d). No fluorescence was detected in $ApoE^{-/-}$ mice that were injected with supernatants from labelled NMVs (Supplementary Fig. 3). Thus, we conclude that NMVs adhere preferentially to atheroprone sites within arteries in conditions of hypercholesterolaemia.

### Oscillatory shear stress promotes NMV adhesion to HCAEC.
In order to investigate the mechanism of preferential adhesion of NMVs to atheroprone endothelium, we carried out in vitro experiments to quantify adhesion of NMVs to HCAEC cultured under different flow parameters. We used oscillatory shear stress (OSS) to model flow at atheroprone regions and high shear stress (HSS) to model the flow found at atheroprotected areas and human NMVs were added to the perfusion media for 2 h after a 72 h conditioning period. As with our in vivo data, we found that under flow conditions, more NMVs adhered to HCAEC that had been cultured under OSS compared to HSS (Fig. 3a, b). As NMVs have previously been shown to adhere to endothelial cells via a CD18-dependent mechanism[28], we confirmed the presence of

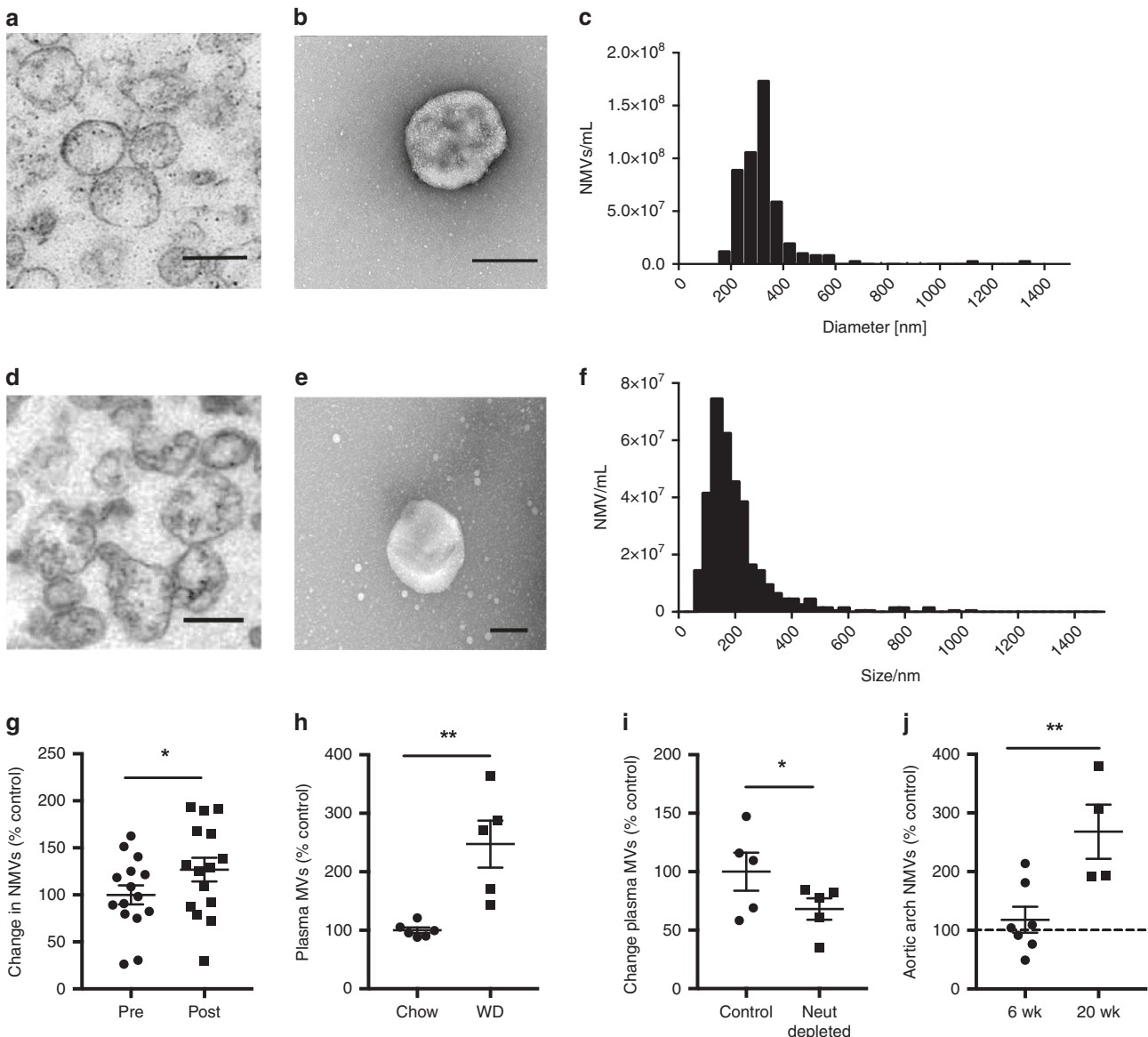

**Fig. 1 Hypercholesterolaemia promotes NMV release and accumulation in the vessel wall.** NMVs were prepared from stimulated human (**a–c**) or mouse (**d–f**) neutrophils. **a**, **d** Transmission electron micrograph of an NMV pellet. **b**, **e** Transmission electron micrograph of a negatively stained NMV sample on carbon-coated copper grids. Magnification of both micrographs ×28,500. Scale bars = 0.25 μm (**a**), 0.2 μm (**b**), 0.25 μm (**c**), 0.2 μm (**d**). Representative histogram showing size distribution of human (**c**) and mouse (**f**) NMVs analysed using Tunable Resistive Pulse Sensing and Nanoparticle Tracking Analysis, respectively. **g** NMVs were detected in human plasma samples before and after high-fat diet using flow cytometry by staining with FITC-anti-CD66b ($n = 15$). **h** Total plasma MVs in $ApoE^{-/-}$ mice fed chow ($n = 6$) or high-fat diet for 6 weeks ($n = 5$) and **i** in $ApoE^{-/-}$ mice fed high-fat diet for 6 weeks with and without neutrophil depletion ($n = 5$) were quantified by flow cytometry. Numbers were normalised to the mean of control samples (filled circles, **g-i**). **j** NMVs were detected in aortic arch homogenates by staining with FITC-anti-mouse Ly6G. The number of NMVs in the aortic arch of $ApoE^{-/-}$ mice on western diet for 6 ($n = 7$) or 20 weeks ($n = 4$) was compared to that of $ApoE^{-/-}$ mice on chow (dotted line) using flow cytometry. Data are presented as mean ± SEM and statistical significance evaluated using a paired (**g**) or unpaired (**h-j**) t-test. *$P < 0.05$, **$P < 0.01$. All n numbers represent independent participants/animals. Source data are provided as a Source Data file.

adhesion molecules on the surface of NMVs as previously described[28–30] (Supplementary Fig. 4a). CD18 expression was detected on all NMVs regardless of the stimulus used to induce their release (Supplementary Fig. 4b). We therefore hypothesised that the preferential adhesion to atheroprone regions may be due to alterations in endothelial expression of the CD18 counter-receptor, intercellular adhesion molecule-1 (ICAM-1). Indeed, HCAEC cultured under oscillatory flow expressed higher levels of ICAM-1 on their surface compared to cells exposed to high shear (Fig. 3c). Consequently, pretreatment of HCAEC cultured under oscillatory shear stress with anti-ICAM-1 antibody significantly

inhibited NMV adhesion (Fig. 3d, e) indicating the preferential adhesion of NMVs to HCAEC exposed to OSS was via an ICAM-1-dependent mechanism. Although PSGL-1 was detected on the surface of NMVs (Supplementary Fig. 4a), HCAEC levels of P-selectin were found to be unchanged by shear stress (Supplementary Fig. 4c) and therefore not likely to mediate the preferential adhesion of NMVs to atheroprone regions.

**NMVs enhance monocyte transendothelial migration**. To determine the functional consequences of NMV interaction with

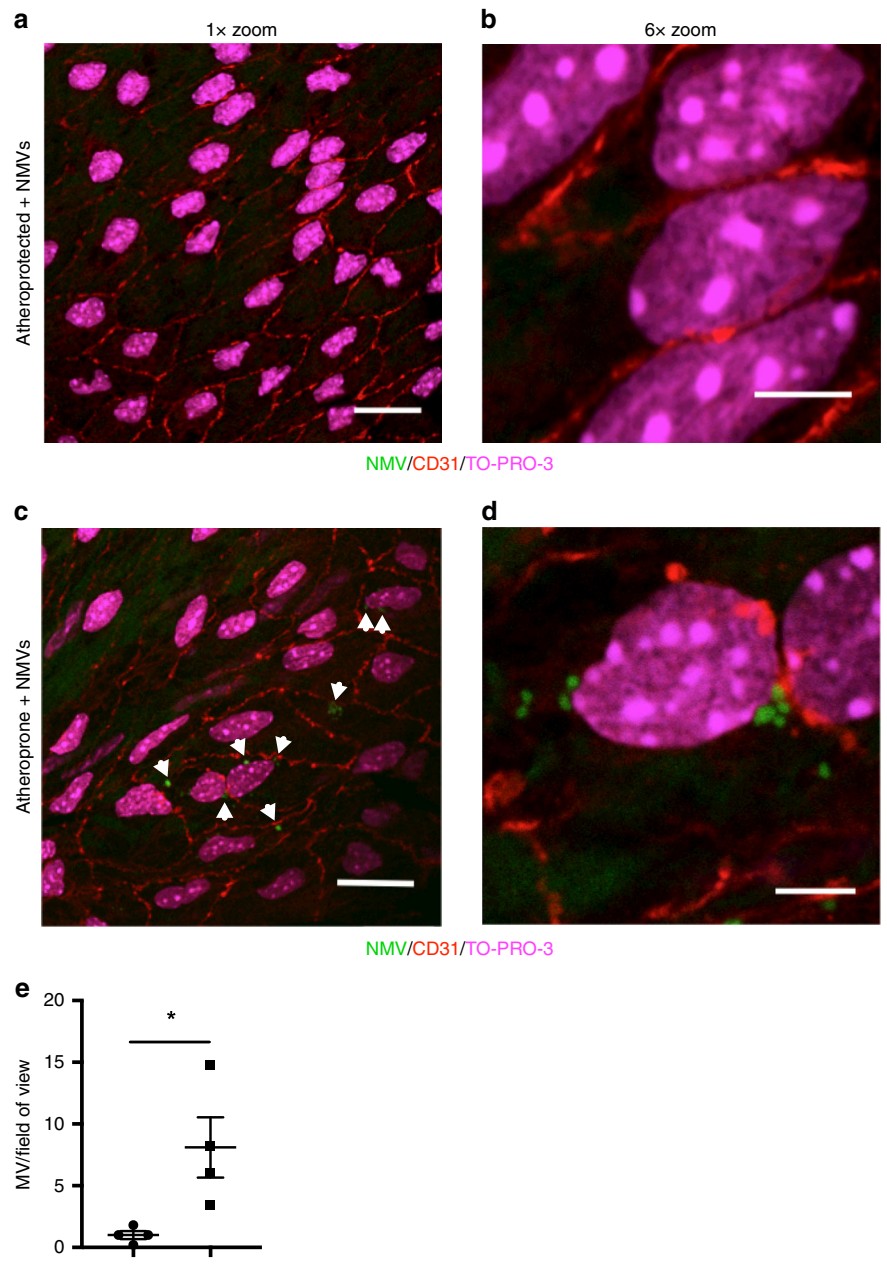

**Fig. 2 NMVs preferentially adhere to atheroprone regions in vivo.** Fluorescently labelled NMVs (green) were injected via the tail vein into *ApoE*$^{-/-}$ mice that had been fed a Western diet for 6 weeks. After 2 h, mice were culled and en face immunostaining of the mouse aortic arch was performed. Representative en face images of NMV adhesion in atheroprotected (outer curvature, **a**, **b**) and atheroprone (inner curvature **c**, **d**) regions of the aorta, visualised by confocal fluorescence microscopy. Endothelial cells were identified by staining with anti-CD31 antibody (red) and cell nuclei were identified using TO-PRO Iodide (magenta). Outer and inner curvature of the ascending aorta were identified by anatomical landmarks and confirmed by characterising the phenotype of endothelial cells; those at the outer curvature were aligned, elongated and uniform—a characteristic of cells under high shear, whereas cells in the inner curvature had a disorganised appearance. Samples were visualised using a ×100 objective at ×1 zoom (**a**, **c**) and at ×6 zoom (**b**, **d**). Scale bars = 20 μm (**a**, **c**), 5 μm (**b**, **d**). **e** Quantification of adherent NMVs presented as mean ± SEM (*n* = 4 animals) and statistical significance evaluated using a paired *t*-test test. *$P < 0.05$. Source data are provided as a Source Data file.

arterial endothelial cells, we investigated their potential effects on pathophysiological processes underpinning vascular inflammation and atherogenesis. In vitro experiments showed that the presence of NMVs significantly enhanced monocyte adhesion to HCAEC under OSS (Fig. 3f, g). Furthermore, monocyte transendothelial migration toward CCL2 was increased in the presence of NMVs in a manner that was dependent on the number of NMV present (Fig. 3h). Crucially, NMVs released from unstimulated neutrophils were unable to induce this increase in

monocyte transmigration (Supplementary Fig. 5a), suggesting that this response is dependent on the properties of NMVs released from stimulated neutrophils. The effect was, however, dependent on the presence of endothelial cells, as NMVs did not influence monocyte migration in the absence of HCAEC (Fig. 3i). Pre-blocking CD18 on the surface of NMVs significantly reduced monocyte transendothelial migration (Fig. 3j). We noted that relatively few fluorescently labelled NMVs adhered to monocytes (Supplementary Fig. 5b) and also NMVs did not activate

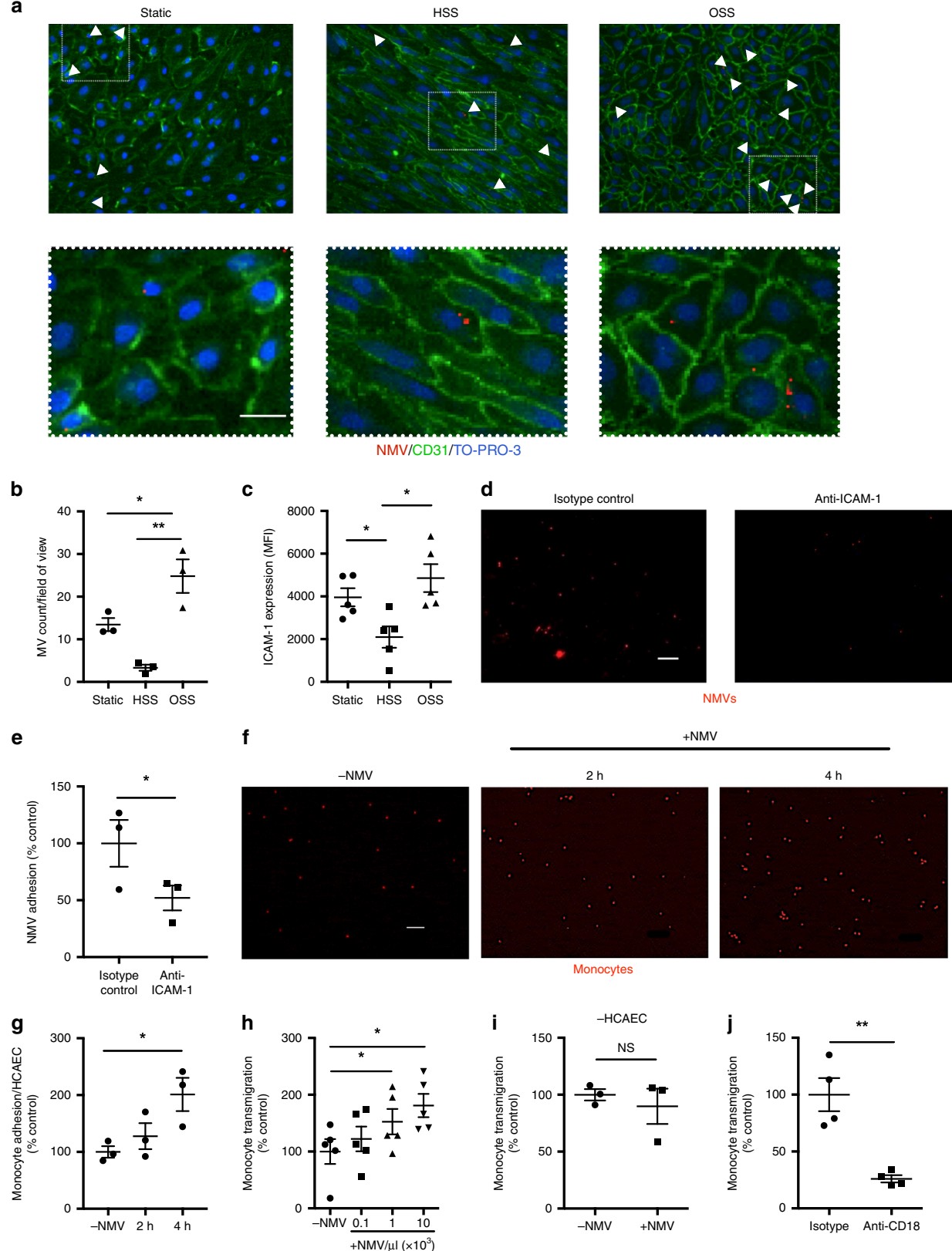

monocytes (as demonstrated by a lack of L-selectin shedding and no induction of CD11b expression; Supplementary Fig. 5c). Thus we conclude that interaction of NMVs released from stimulated neutrophils with endothelial cells, via CD18 on the NMV surface, promotes the subsequent recruitment of monocytes, and that this process does not involve monocyte–NMV interactions. Therefore, we next investigated whether NMVs promote monocyte recruitment by altering endothelial cell inflammatory activation.

**NMVs induce inflammatory activation of HCAEC via RelA.** Given the central role of cytokines and adhesion molecules in

**Fig. 3 NMVs adhere to HCAEC under flow and enhance monocyte migration. a, b** HCAEC were cultured (72 h) under static, high laminar (HSS) or low oscillatory shear stress (OSS). Fluorescently labelled NMVs (red) were added for 2 h. Cells co-stained for CD31 (green) and TO-PRO-3 (blue). **a** Representative images (scale bar = 50 μm) and **b** quantification of NMV adhesion (n = 3). **c** HCAEC cultured as for **a** were incubated with fluorescently labelled anti-ICAM-1, changes analysed by flow cytometry and displayed as mean fluorescence intensity (MFI) (n = 5). **d, e** HCAEC cultured under OSS (72 h) were incubated with anti-ICAM-1 or isotype control antibody (100 μg mL$^{-1}$) for 1 h before addition of fluorescently labelled NMVs. **d** Representative images (scale bar = 100 μm) and **e** quantification of NMV (red) adhesion. Data expressed as percentage of mean of isotype sample (n = 3). **f, g** After 72 h exposure to OSS, NMVs were perfused over conditioned HCAEC for 2 h. Fluorescently labelled monocytes (red) were added to perfusion media for 2–4 h. **f** Representative images (scale bar = 100 μm) and **g** quantification of adherent monocytes (n = 3). Data expressed as percentage of mean number of adherent monocytes per HCAEC in control samples (−NMV). **h–j** HCAEC cultured on transwells were incubated with (+NMV) or without (−NMV) NMVs for 30 min followed by addition of monocytes. Monocyte transmigration toward CCL2 (5 nmol/L) was measured after 90 min (n = 5). **i** Monocyte transmigration was repeated in absence of HCAEC (n = 3). **j** NMVs were treated with anti-CD18 or isotype control for 20 min, washed and the transmigration experiment repeated (n = 4). **h–j** Data expressed as percentage of mean of control samples (−NMV/isotype) and presented as mean ± SEM. Statistical significance evaluated using a paired t-test (**e, i, j**) or one-way ANOVA followed by Tukey's (**b, c**) or Dunnett's (**g, h**) test. NS not statistically significant, *P < 0.05, **P < 0.01. All n numbers represent independent experiments. Source data are provided in Source Data file.

monocyte recruitment, we investigated the effects of NMVs on endothelial expression of these inflammatory factors. Following 2 or 4 h incubation with NMVs under static conditions, HCAEC showed a significant increase in the release of the monocyte chemoattractant CCL2 and the neutrophil chemoattractant CXCL8 compared to HCAEC alone, whereas levels of IL-6 were not significantly increased (Fig. 4a). The increase in CCL2 release at 4 h is physiologically relevant since it is comparable to that measured when HCAEC were incubated with a well-described inflammatory stimulus, tumour necrosis factor (TNF) (1154 ± 298 pg mL$^{-1}$ (mean ± SEM). We then repeated the 4 h incubation adding NMVs under OSS conditions as disturbed flow itself is known to be pro-inflammatory[27,31]. NMVs induced a significant increase in the release of IL-6 and CXCL8 from HCAEC under OSS (Fig. 4b). In addition, NMVs induced an increase of ICAM-1, vascular cell adhesion molecule-1 (VCAM-1), and CCL2 protein levels under static conditions (Fig. 4c) in HCAEC. These cytokines and adhesion molecules were not detectable in NMVs alone indicating that their production by HCAEC can be induced by NMVs. Gene expression changes in *ICAM-1* and *VCAM-1* were observed in HCAEC incubated with NMVs both under static and OSS conditions after 2 h, whereas increased levels of *CCL2* were only observed under static conditions (Fig. 4d). Additionally, the increase in gene expression was not due to contaminants contained within the NMV supernatants (Supplementary Fig. 6). Importantly, NMVs released by stimulated neutrophils were able to induce significantly greater increases in gene expression in HCAECs than those from unstimulated cells (Supplementary Fig. 7a) suggesting that NMVs released from stimulated neutrophils are able to induce greater endothelial cell activation.

To interrogate the mechanism by which NMVs induced increased expression of inflammatory molecules, we investigated whether they influence the NF-κB pathway, which is a central regulator of inflammation in arterial endothelial cells. We focused on RELA, an abundant proinflammatory NF-κB subunit in endothelial cells, and found that NMVs released by activated neutrophils induced *RELA* expression in HCAEC after 2 h under both static and OSS conditions (Fig. 4e) unlike those released by unstimulated cells (Supplementary Fig. 7b). We hypothesised that this was mediated via transfer of NMV cargo to endothelial cells and inflammatory activation via increased RELA expression. Consistent with this, we found that NMVs were internalised by arterial endothelial cells, both in vitro (Fig. 5a) and in vivo (Fig. 5b, c). Targeted labelling of early endosomes was used to determine the localisation of NMVs within the cytoplasm and some NMVs were found to colocalise with this marker (Fig. 5a). No signal in the NMV channel was detected in cells that were treated with supernatants from labelled NMVs where F-actin was

labelled (Supplementary Fig. 8). This was a metabolically active process (i.e. endocytosis/macropinocytosis rather than diffusion) as internalisation was significantly reduced when cells were incubated at 4 °C or room temperature compared to 37 °C (Fig. 5d). Furthermore, TNF, a known inducer of ICAM-1 expression, increased internalisation whereas blocking ICAM-1 with anti-ICAM antibody partially inhibited NMV internalisation (Fig. 5e), both of which support a role for ICAM-1. Consistent with this, increased NMV internalisation was observed when HCAEC were cultured under OSS compared to static conditions (Fig. 5f).

**NMVs induce NF-κB activation in HCAEC via delivery of miR-155.** We next investigated whether NMVs contained miRNAs that could influence gene expression in recipient endothelial cells. We focussed on miRNAs that are found in activated neutrophils and are not found constitutively in endothelial cells. RT-qPCR analysis revealed that NMVs contained several miRNAs, including regulators of inflammation (*miR-9, miR-150, miR-155, miR-186, miR-223*; Fig. 6a).

We next determined whether plasma MV expression of the most abundant miRNAs, *miR-223* and *miR-155*, were altered by high-fat diet in human subjects and found that only *miR-155* expression levels were significantly elevated (Fig. 6b). We compared the relative levels of expression of *miR-155* in NMVs derived from unstimulated neutrophils and neutrophils exposed to fMLP or modified lipoprotein. NMVs from fMLP and acLDL stimulated neutrophils had relatively more *miR-155* expression than those from unstimulated neutrophils (Supplementary Fig. 9). Moreover, stimulation of neutrophils with fMLP or acLDL resulted in a significant increase in NMV release compared to that from unstimulated neutrophils (Supplementary Fig. 10). It was therefore concluded that stimulated neutrophils release greater number of NMVs with increased miRNA content. In mice, *miR-155* expression levels were found to be increased in plasma MVs and NMVs after Western diet compared to chow (Fig. 6c, d). Additionally, following Western diet feeding for 6 weeks, NMVs had comparatively higher levels of *miR-155* expression than plasma NMVs as a whole (copy number for plasma MVs after WD = 171 ± 18.1 (mean ± SEM); copy number for NMVs after WD = 416 ± 13.8 (mean ± SEM)). *miR-155* has been shown to increase NF-κB expression by targeting its negative regulator, BCL6[32–34]. We therefore hypothesised that NMVs activate NF-κB by delivering *miR-155*, which reduces *BCL6* expression. Consistent with this, incubation of HCAEC with NMVs led to enhanced endothelial expression of *miR-155* associated with reduced *BCL6* expression (Fig. 6e). Exposure to high-fat diet for 1 week augmented the ability of NMVs to increase *miR-155*

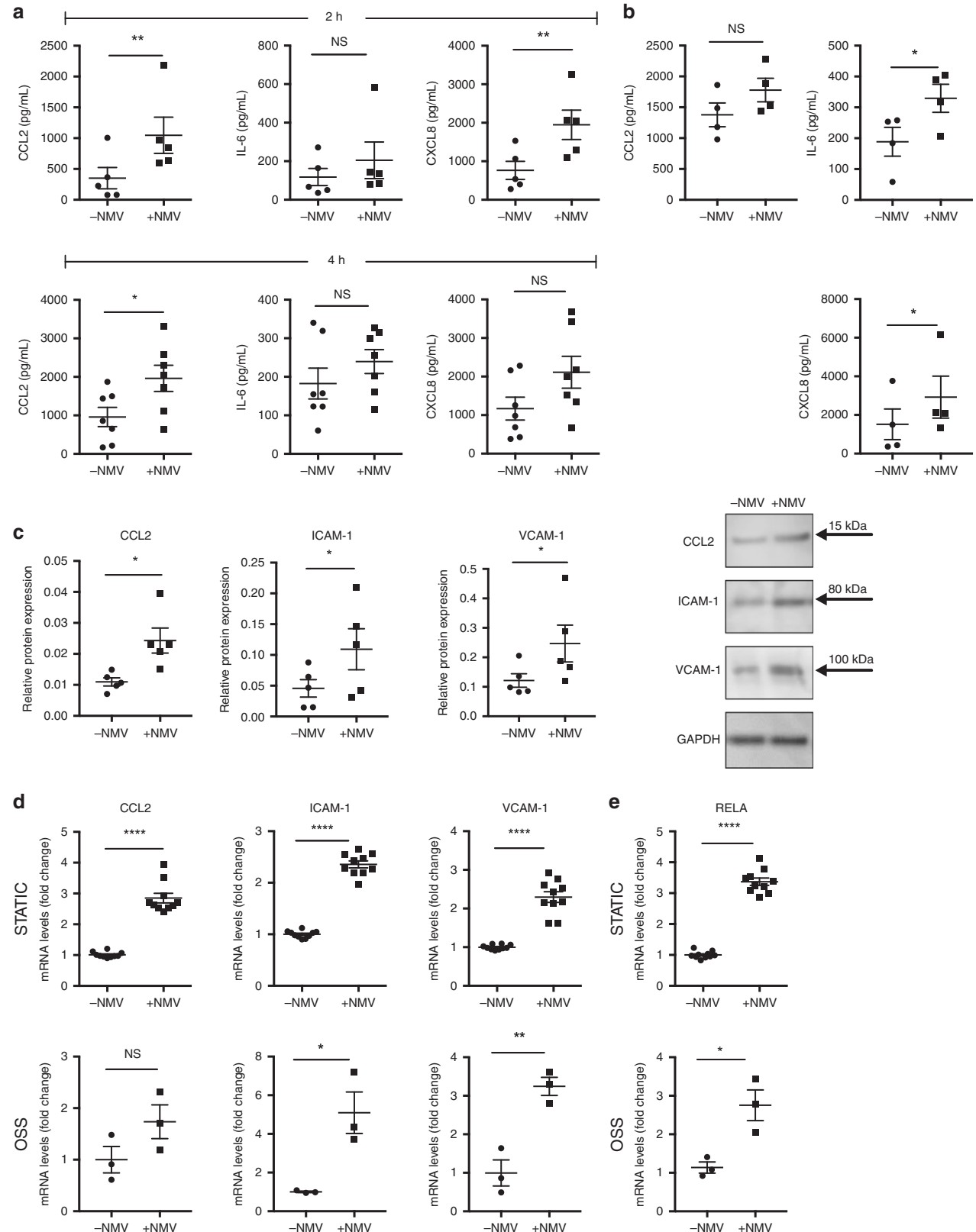

expression levels in HCAEC (Fig. 6f). The potential of MVs to deliver miR-155 to EC was confirmed by demonstrating that NMVs from wild-type mice could enhance *miR-155* expression in HCAEC whereas NMVs from *miR-155*$^{-/-}$ mice could not (Fig. 6g). Indeed, we found a small but statistically significant reduction in *miR155* copy number in HCAEC when incubated

with *miR155*$^{-/-}$ NMVs, although the mechanism for this unexpected finding are unknown and could be due to levels being near the lower detection limit of the assay. The reduced expression of *BCL6* seen with NMV incubation was reversed in endothelial cells transfected with an antagomir that blocks *miR-155* function, indicating that BCL6 is negatively regulated by

**Fig. 4 HCAEC activation by NMVs.** HCAEC were cultured for 72 h under static conditions (**a**) or oscillatory shear stress (OSS; **b**) and were then incubated with (+NMV) or without (−NMV) NMVs for 2 h (**a**; n = 5) and 4 h (**a**; n = 7, **b**; n = 4). Release of CCL2, CXCL8 and IL-6 into the media was analysed using cytometric bead array (**a**) or ELISA (**b**). **c** HCAEC were incubated with (+NMV) or without (−NMV) NMVs for 2 h under static conditions and alterations in inflammatory protein (n = 5) expression were investigated using western blotting. Samples were quantified using densitometry and normalised to GAPDH. **d, e** HCAEC were incubated with (+NMV) or without (−NMV) NMVs for 2 h under static conditions (n = 10; upper panel) or OSS (n = 3; lower panel) and gene expression changes measured using RT-qPCR. Samples were normalised to β-actin. Results are presented as mean ± SEM and statistical significance evaluated using a paired t-test. NS not statistically significant, *P < 0.05, **P < 0.01, ***P < 0.001. All n numbers represent independent experiments. Source data are provided as a Source Data file.

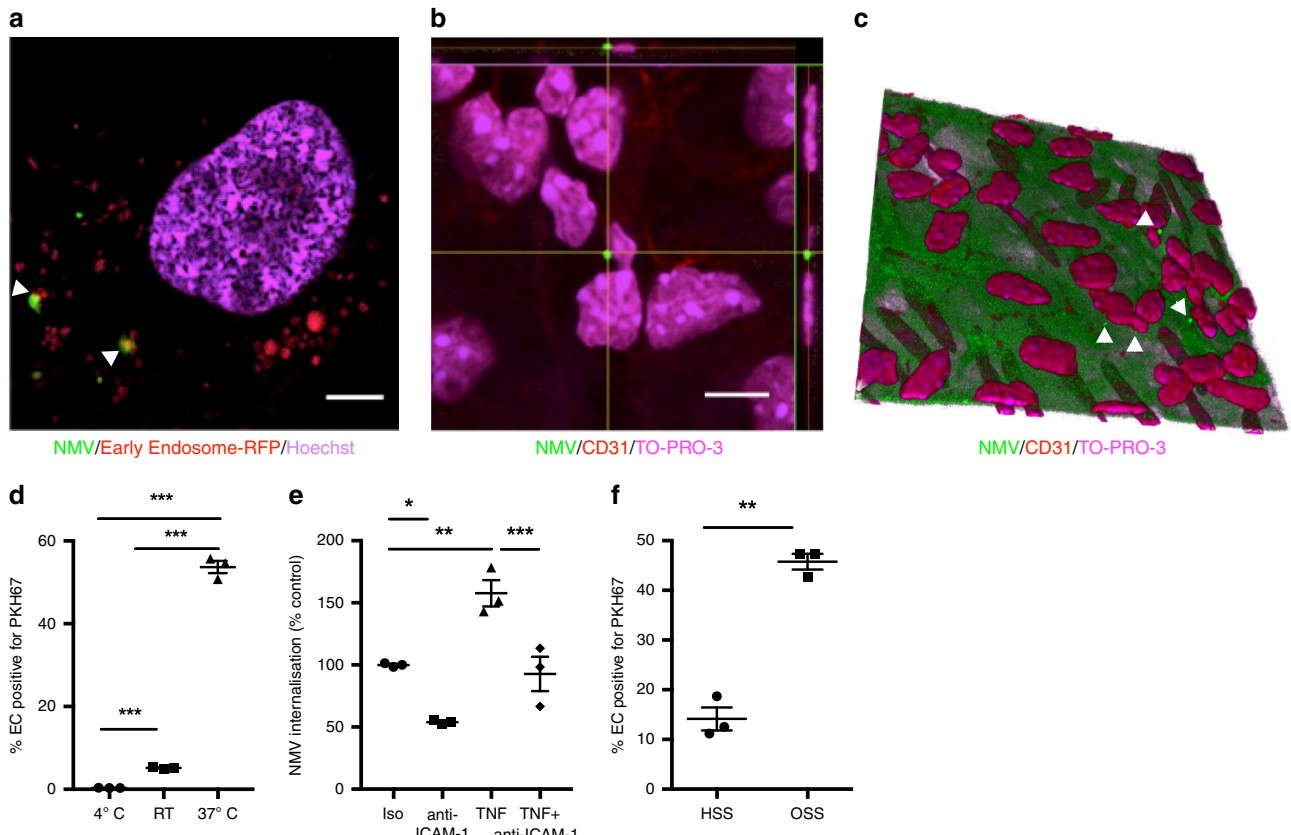

**Fig. 5 NMV internalisation by endothelial cells. a** Confocal z stack image taken at a depth of 0.8 μm from the base of an HCAEC incubated with fluorescently labelled NMVs (green) and labelled with early endosome marker (CellLight® Early Endosomes-RFP; red) and Hoechst (nuclei; blue). The arrowheads indicate colocalisation of NMV and CellLight® Early Endosomes-RFP fluorescence. Scale bar = 5 μm. **b** Orthogonal view (scale bar = 10 μm and **c** 3D reconstruction of an ApoE⁻/⁻ mouse aorta stained en face with anti-CD31 (endothelial cells; red) and TO-PRO-3 (nuclei; magenta) showing internalisation of fluorescently labelled NMVs (green) 2 h after i.v. injection. Elastin autofluorescence also appears as green. CD31 expression on the apical surface was used for orientation and the plane of view set just below. Note the misaligned endothelial cell nuclei, characteristic of an area of disturbed flow. Arrows denote NMVs. HCAEC were cultured under static conditions (**d, e**) or flow conditions (**f**) for 72 h followed by incubation with fluorescently labelled NMVs for 2 h under static (**d, e**) or flow (**f**) conditions (n = 3). Fluorescence from residual surface bound NMVs was quenched with trypan blue and data were analysed for changes in mean fluorescence intensity by flow cytometry. **d** The experiment was performed at 4 °C, room temperature (RT) or 37 °C. **e** HCAEC were incubated at 37 °C in the presence of TNF (4 h prior to the addition of NMVs) and/or anti-ICAM-1 or isotype control. Data are expressed as a percentage of the mean of the isotype control samples. Data are presented as mean ± SEM and statistical significance evaluated using one-way ANOVA followed by Tukey's post test for multiple comparisons (**d, e**) or paired t-test (**f**). *P < 0.05, **P < 0.01, ***P < 0.001. All n numbers represent independent experiments. Source data are provided as a Source Data file.

miR-155. Consequently, the increase in expression of RELA and its downstream target genes VCAM-1, ICAM-1 and CCL2 induced by NMVs was significantly decreased in the presence of miR-155 antagomir (Fig. 6h). In support of this, injection of NMVs into ApoE⁻/⁻ mice induced a significant increase in arterial expression of miR-155 (Fig. 7a) and a subsequent reduction in BCL6 expression (Fig. 7b, c).

**NMVs enhance plaque formation in a miR-155 dependent manner.** Having shown regulation of RelA expression in vitro, we

investigated whether circulating NMVs could induce focal inflammation in vivo. To investigate the function of NMVs in vivo, ApoE⁻/⁻ mice were injected twice weekly for 6 weeks with NMVs to chronically increase circulating NMV levels by ~30%, similar to the increase in circulating NMVs observed in human subjects after 7 days on an atherogenic diet (Fig. 1g). Levels of RELA in the aorta were assessed by en face confocal microscopy and found to be markedly and selectively increased at atheroprone regions in response to NMV injection (Fig. 8a, b). Under these conditions, RELA localised partially to the nucleus and a proportion located to

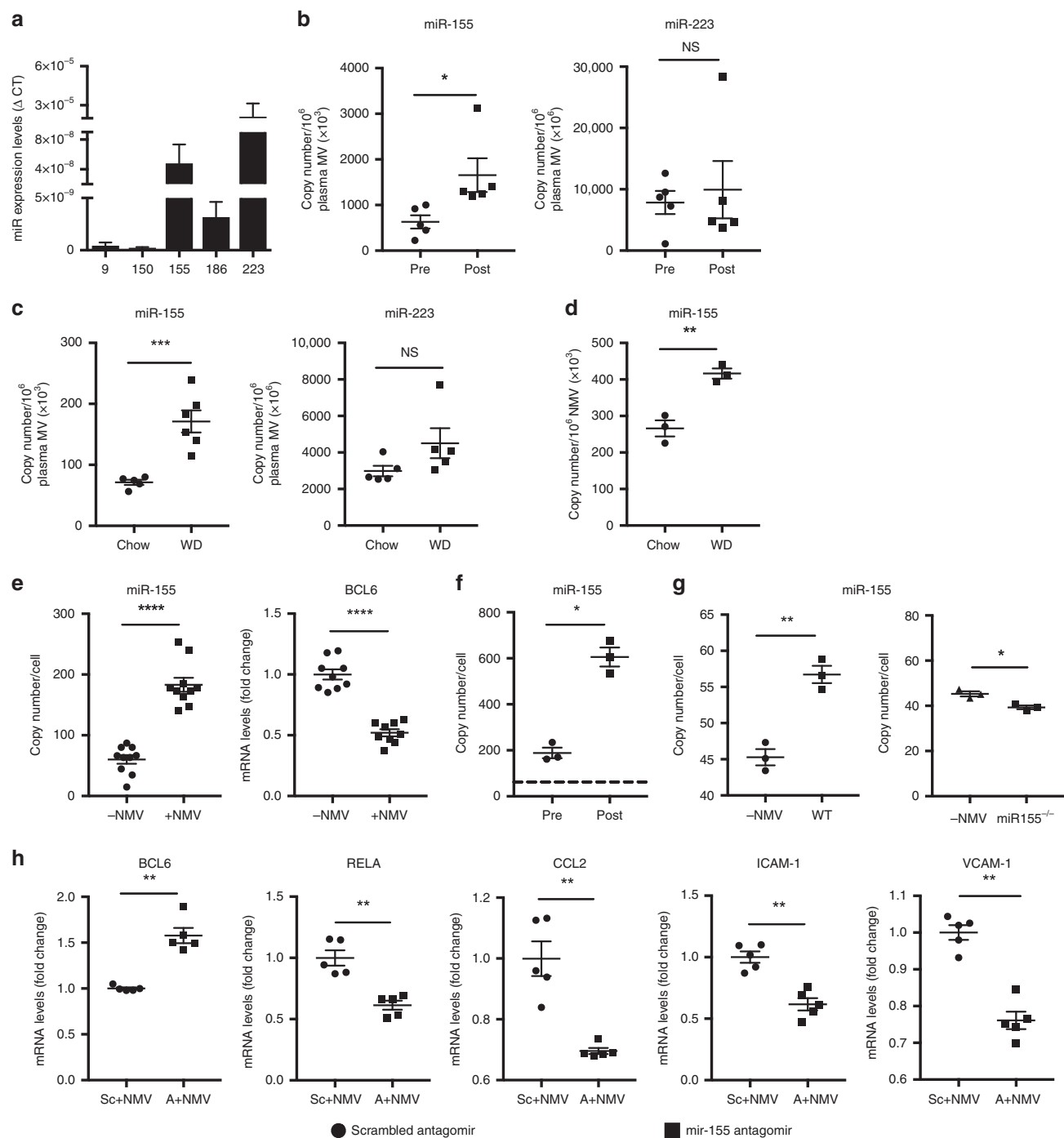

**Fig. 6 NMVs contain miRNAs that are delivered to HCAEC and alter gene expression. a** miRNA content of NMVs prepared from stimulated human neutrophils quantified by RT-qPCR ($n = 5$). **b** *miR-155* and *miR-223* content of MVs in human plasma pre- and post high-fat diet (HFD, $n = 5$), or (**c**) plasma from mice fed chow or Western diet (WD, 6 weeks; $n = 5$) was quantified by RT-qPCR. **d** *miR-155* expression levels in NMVs isolated from mice fed chow or Western diet ($n = 3$) was measured by RT-qPCR. **e** HCAEC were incubated with (+NMV) or without (−NMV) NMVs for 2 h and *miR-155* ($n = 10$) and *BCL6* ($n = 9$) expression levels measured by RT-qPCR. **f** HCAEC were incubated with NMVs prepared from human neutrophils isolated pre- and post HFD and *miR-155* expression levels quantified by RT-qPCR ($n = 3$). The dotted line shows the mean copy number per HCAEC in the absence of NMVs. **g** HCAEC were incubated with NMVs prepared from *miR-155*$^{-/-}$ vs. wild type mouse neutrophils and *miR-155* expression levels quantified by RT-qPCR ($n = 3$). **h** HCAEC were transfected with 25 pmol of *miR-155* antagomir (A) or scrambled control (Sc) and incubated with NMVs for 2 h. HCAEC expression of *BCL6*, *RELA* and its downstream targets was investigated by RT-qPCR ($n = 5$). Data are presented as mean ± SEM and statistical significance evaluated using a paired or unpaired *t*-test as appropriate. NS not statistically significant, *$P < 0.05$, **$P < 0.01$, ***$P < 0.001$, ****$P < 0.0001$. All *n* numbers represent independent experiments. Source data are provided as a Source Data file.

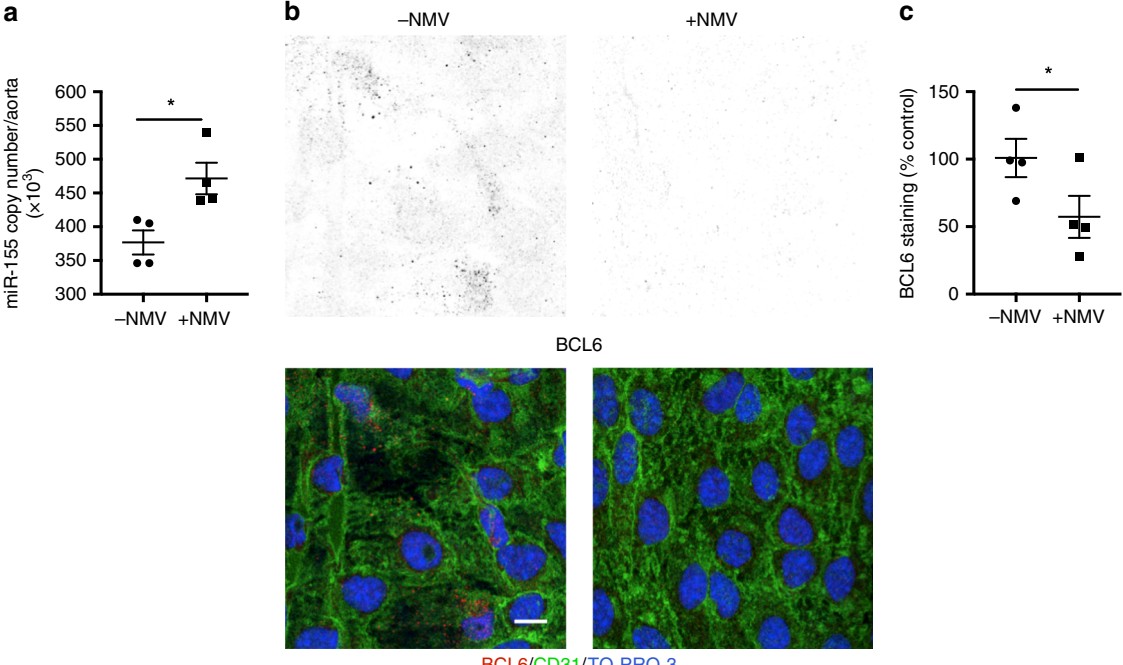

**Fig. 7 NMVs increase miR-155 and reduce BCL6 expression in atheroprone regions in vivo. a** $ApoE^{-/-}$ mice were injected with NMVs via the tail vein and *miR-155* expression levels in the aorta were quantified by RT-qPCR 2 h after injection of saline (−NMV) or NMVs (+NMV; $n = 4$). All RT-qPCR samples were normalised with β-actin. **b**, **c** Carotid arteries were isolated from $ApoE^{-/-}$ mice and incubated ex vivo with (+NMV) or without (−NMV) NMVs for 2 h and stained en face with anti-BCL6 antibody (black in single channel, upper panels/red in merged images, lower panels). Endothelial cells were identified by staining with anti-CD31 (green) and cell nuclei were identified using TO-PRO-3 Iodide (blue). Representative en face images of the BCL6 channel and merged channels are shown. Scale bar = 20 μm. Total fluorescence intensity of BCL6 expression was quantified using ImageJ software ($n = 4$) and expressed as a percentage of the mean fluorescence in the control samples (−NMV). Data are presented as mean ± SEM and statistical significance evaluated using an unpaired *t*-test. *$P < 0.05$. All *n* numbers represent independent animals. Source data are provided as a Source Data file.

the cytoplasm, suggesting that NMVs induce partial activation of NF-κB at atheroprone sites, and also prime endothelial cells for enhanced inflammatory responses by increasing total NF-κB expression.

Having determined that NMVs preferentially adhere to atheroprone regions, enhance monocyte transendothelial migration and regulate RELA, we hypothesised that this could lead to enhanced plaque formation. Consistent with this hypothesis, en face Oil Red O staining revealed significantly more atherosclerotic plaque formation in mice treated with NMVs (twice weekly injections over a 6-week period as for the RELA expression experiments above) compared to those treated with saline (Fig. 8c, d). En face staining with MAC3 antibody revealed enhanced recruitment of monocytes/macrophages in response to NMV injection (Fig. 8e, f). Thus, we conclude that systemic NMVs can induce focal activation of NF-κB at atheroprone sites and, thereby, amplify vascular inflammation and accelerate lesion formation.

To further elucidate the role of *miR-155* in NMV-induced enhancement of atherosclerotic plaque formation, wild type or *miR-155^-/-* mouse NMVs were injected twice weekly for two weeks into $ApoE^{-/-}$ mice that had been on Western diet for 4 weeks. Wild type NMVs significantly increased atherosclerotic plaque area in the aortic root compared to control (no NMVs injected), whereas NMVs isolated from *miR-155^-/-* mice had no significant effect (Fig. 9a) as determined by Oil Red O staining. Additionally, immunohistochemical analysis of MAC3 staining in the aortic root revealed that significantly higher levels of monocyte/macrophages were detected in atherosclerotic plaques in $ApoE^{-/-}$ mice that were injected with wild type NMVs compared to control (Fig. 9b). However, *miR-155^-/-* NMVs did not affect monocyte recruitment and significantly fewer were detected in plaques compared to wild

type NMV injected mice. It was therefore concluded that NMVs enhance vascular inflammation and atherosclerosis and that *miR-155* is essential for this pathogenic process.

## Discussion

Neutrophils are the most abundant leucocyte in human circulation and are essential for an effective innate immune response. There is also increasing evidence for their role in atherosclerosis. High fat feeding, both in humans and in mouse models, increases the level and activation of neutrophils[35–37]. Although small numbers of neutrophils have been detected within the core of developing atherosclerotic plaques, this peaks at early stages (4 weeks after high-fat diet for $ApoE^{-/-}$ and 6 weeks for $LDL^{-/-}$ models of experimental atherosclerosis) and neutrophils are rarely observed beyond these time points[37,38]. Nevertheless, there is evidence to suggest that neutrophils may play a role in plaque development through mechanisms that do not require them to be present within the plaque core, such as neutrophil extracellular trap formation (NETosis) in response to the presence of cholesterol crystals[39]. In addition to NETosis, activated neutrophils are known to release MVs[29,40,41]. However, a role for NMVs in atherosclerosis has not previously been investigated. Here we show that: (i) high-fat diet raises levels of circulating NMVs; (ii) NMVs adhere preferentially to atheroprone regions; (iii) once adherent, NMVs are internalised by endothelial cells and deliver *miR-155* and that (iv) NMVs from stimulated neutrophils activate endothelial cells, enhance monocyte recruitment and exacerbate atherosclerotic plaque formation in a *miR-155*-dependent manner (see Fig. 10 for proposed molecular model).

We observed that a high-fat diet, a known driver of atherosclerosis[42,43], potently enhanced circulating levels of NMVs

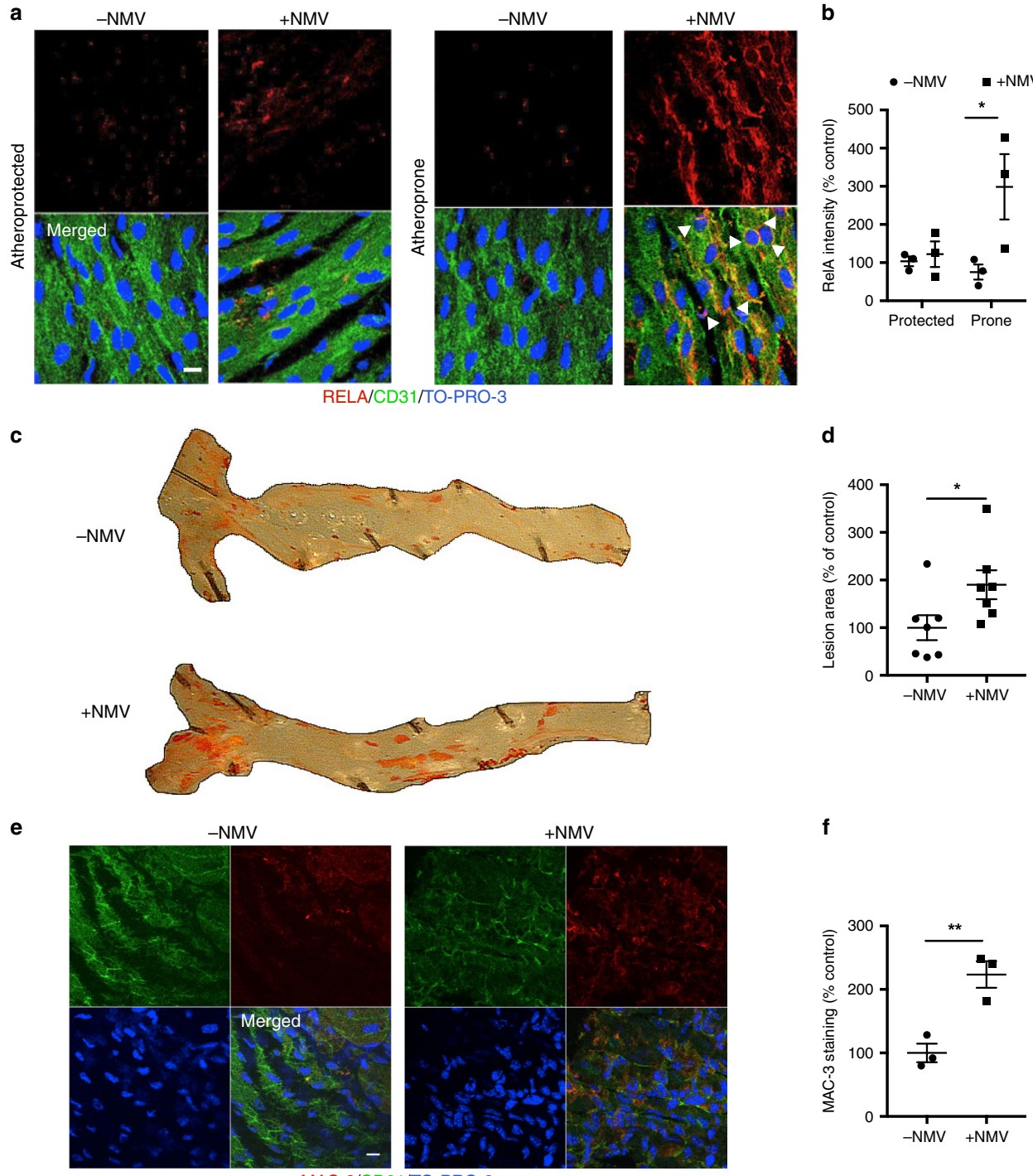

**Fig. 8 NMVs induce activation of NF-κB in atheroprone regions and enhance atherosclerosis.** *ApoE⁻/⁻* mice fed a Western diet were injected with NMVs, or an equivalent volume of saline, twice weekly via the tail vein for 6 weeks. **a** Representative en face images of RELA (red) expression in the aorta of mice injected with saline (−NMV) or NMVs (+NMV) in atheroprone regions visualised using confocal fluorescence microscopy. Scale bar = 10 μm. Endothelial cells were identified by staining with anti-CD31 (green) and cell nuclei were identified using TO-PRO-3 Iodide (blue). Examples of nuclear staining of RelA indicated with arrowheads. **b** Mean fluorescence intensity was quantified using ImageJ software and data expressed as mean ± SEM (*n* = 5). **c, d** Plaque formation was measured in dissected aortae using en face Oil Red O staining and imaged by bright field microscopy. **c** Representative images are shown. **d** Areas of plaque formation were determined in the entire aorta using NIS-elements analysis software (*n* = 7). **e, f** The aortic arches of mice injected with saline (−NMV) or NMVs (+NMV) were studied by en face staining to quantify macrophages (MAC-3, red). **e** Endothelial cells were identified by staining with anti-CD31 antibody (green) and cell nuclei were identified using TO-PRO-3 Iodide (blue). Scale bar = 10 μm. **f** Mean fluorescence intensity was quantified using Image J software (*n* = 3). Data are expressed as a percentage of the mean of the control samples (−NMV) and presented as mean ± SEM. Statistical significance was evaluated using two-way ANOVA followed by Bonferroni's post hoc test (**b**) or an unpaired *t*-test (**d**, **f**). *$P < 0.05$, **$P < 0.01$. All *n* numbers represent independent animals. Source data are provided as a Source Data file.

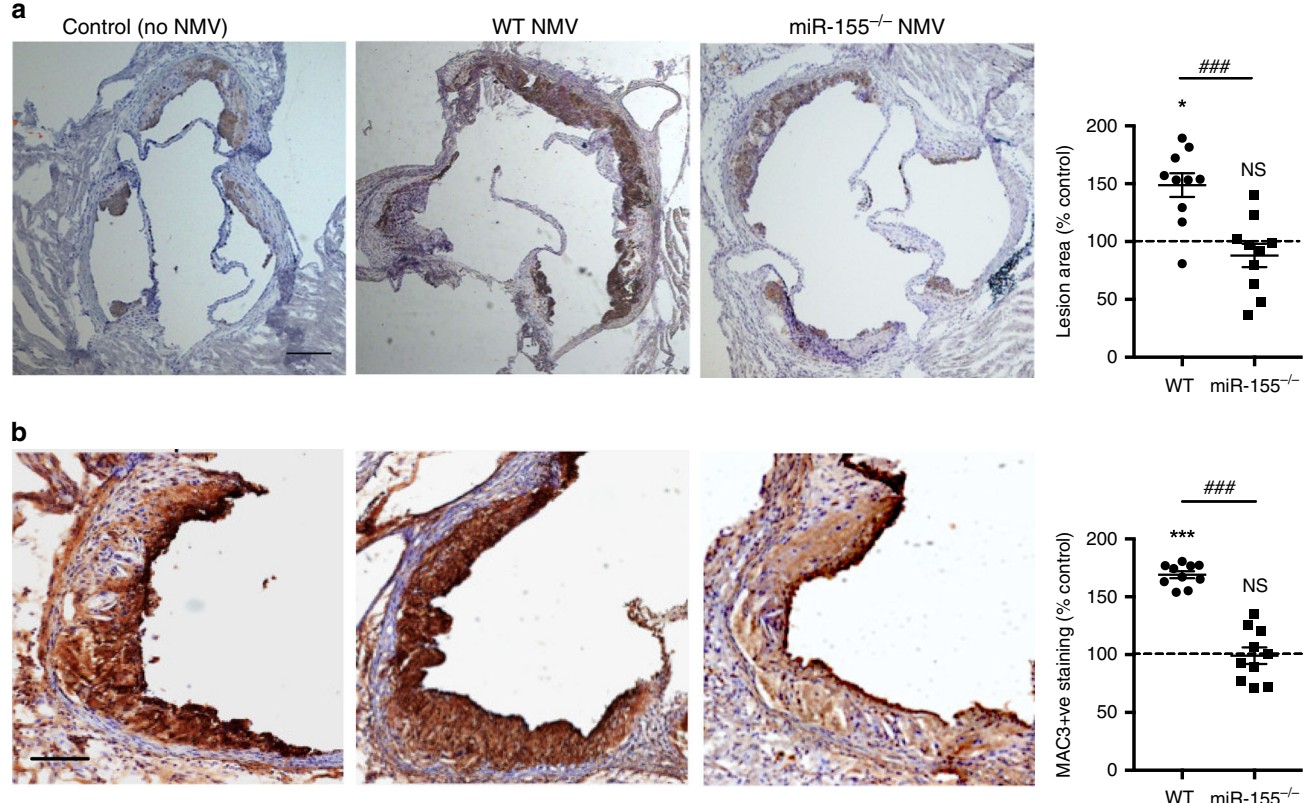

**Fig. 9 NMVs enhance formation and macrophage content of atherosclerotic plaques in a miR-155-dependent manner.** $ApoE^{-/-}$ mice fed a Western diet for 6 weeks were injected with NMVs isolated from wild type (WT) or $miR\text{-}155^{-/-}$ mouse neutrophils, or an equivalent volume of saline (control), twice weekly via the tail vein for the final 2 weeks. **a** Plaque formation and **b** macrophage content was measured in frozen aortic root sections by Oil Red O and MAC-3 staining respectively and imaged by bright field microscopy. **a** Representative images of Oil Red O staining from control, wild type NMV and $miR\text{-}155^{-/-}$ NMV injected $ApoE^{-/-}$ mice are shown. Scale bar = 200 μm. Areas of plaque formation were determined in aortic root sections using NIS-elements analysis software ($n = 10$). **b** Representative images of MAC-3 staining from control, wild type NMV and $miR\text{-}155^{-/-}$ NMV injected $ApoE^{-/-}$ mice are shown. Scale bar = 50 μm. MAC-3 positive staining within plaques was determined using NIS-elements analysis software ($n = 10$). Data are expressed as a percentage of the mean of the control samples (no NMV) and presented as mean ± SEM. Statistical significance was evaluated using one-way ANOVA followed by Tukey's post hoc test. NS not statistically significant, $*P < 0.05$, $***P < 0.001$ compared to control. $^{###}P < 0.001$ wild type NMV compared to $miR\text{-}155^{-/-}$ NMV injected mice. All $n$ numbers represent independent animals. Source data are provided as a Source Data file.

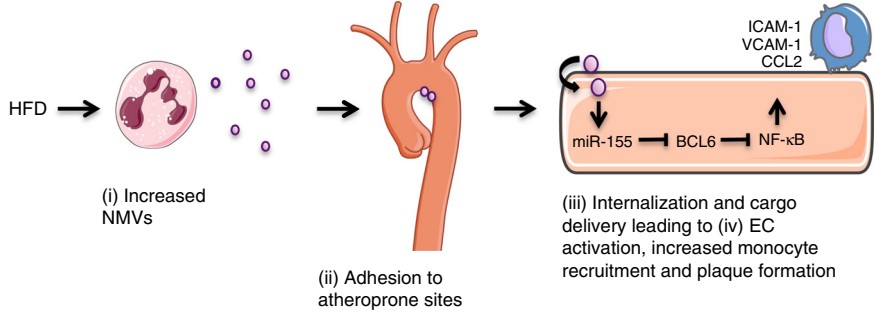

**Fig. 10 Proposed model of the molecular mechanism of action of NMVs in atherosclerosis.** High fat diet increases the level of circulating NMVs (i) that preferentially adhere to atheroprone regions of arteries (ii) where they become internalised by endothelial cells (iii). Delivery of $miR\text{-}155$ to the endothelial cells downregulates BCL6, leading to an increase in NF-κB expression and subsequent inflammatory activation. This results in an increase in the number of monocytes recruited to the vessel wall and enhanced atherosclerotic plaque formation.

in healthy humans. Additionally, we found that modified LDL was able to induce the release of large numbers of MVs from neutrophils with high levels of microRNA, suggesting that dyslipidaemia may induce the increased levels of circulating MVs observed in both our human and murine studies. It is important to note that, despite the differences in the longevity of the different models used

in the current study—a short term increase in fat intake in healthy humans and a longer term high-fat diet in hypercholesterolemic mice - similar changes in plasma MVs were observed with the absolute number rather than the proportions of MVs changing. Our findings in healthy human subjects were mirrored in hypercholesterolemic mice and led to an increase of NMVs in vessel walls.

Although levels of NMVs in the vessel wall increased over time, this could be due to accumulation of NMVs that are not degraded once internalised or, alternatively, it could be due to an increase in the rate of uptake due to exacerbation of local inflammation. The observations from our murine studies are in agreement with findings from Leroyer and colleagues who detected the presence of granulocyte-derived MVs in human atherosclerotic plaques[18]. Further studies into the role of NMVs in patients with cardiovascular disease will need to take into account the effects of current treatments such as statins on MV numbers[44] as well as the effect of comorbidities such as diabetes, which is known to be correlated with increased circulating MVs[45]. However, since our data clearly show a link between NMVs and atherosclerosis, this may warrant further investigation in a large cohort of well-characterised patients at risk of cardiovascular disease.

Atherosclerosis is a focal disease that occurs in regions of disturbed flow. NMV-endothelial cell interactions were increased when under conditions of disturbed flow and at atheroprone sites in mice fed on a high-fat diet. This correlated with the expression of *ICAM-1*, and adhesion was indeed found to be ICAM-1 dependent, similar to findings previously described investigating NMV interactions with HUVEC under static conditions[28]. Interestingly, interaction of NMVs from stimulated neutrophils with endothelial cells further increased *ICAM-1* expression suggesting that this interaction may lead to an augmentation of subsequent NMV-endothelial cell interactions (i.e. positive feedback). This could provide a mechanism by which NMVs accumulate in the vessel wall of mice on a high-fat diet over time. Once adherent, NMVs were internalised by endothelial cells. ICAM-1 was also required for the internalisation of NMVs and, interestingly, Muro et al. have described a pathway for endocytosis involving clustering of ICAM-1[46,47]. It is plausible that NMVs utilise this pathway, providing a mechanism by which increased levels of internalisation occur in areas where there is increased expression of this adhesion molecule, such as atheroprone regions. Further studies to investigate the precise mechanisms by which NMVs are internalised and the fate of NMVs once internalised are of great importance as this could potentially be exploited in the future for targeted delivery of therapeutic agents.

Although we found that unstimulated neutrophils released NMVs albeit in lower numbers, we also found that, unlike NMVs from stimulated neutrophils, these NMVs were unable to induce endothelial cell activation. It is likely that these functional differences are due to divergent cargo as previously described for NMVs released from adherent vs. non-adherent neutrophils[48]. There has been much interest in the ability of extracellular vesicles to transfer genetic material from the parent cell to a target cell[14,15]. Here we demonstrate that NMVs contain numerous miRNAs implicated in atherosclerosis and vascular inflammation, the most abundant of which were *miR-155* and *miR-223*. High fat feeding of both healthy human volunteers and *ApoE*−/− mice resulted in a significant increase in plasma MV and NMV levels of *miR-155* but not *miR-223*. In addition, incubation of HCAECs with NMVs increased cellular expression of *miR-155*, which was further increased when NMVs were prepared from subjects who had undertaken the high-fat diet study. We also found that levels of *miR-155* were greater in NMVs derived from stimulated compared to unstimulated neutrophils. We concluded that *miR-155* was delivered to, but not induced in, HCAEC as NMVs isolated from *miR-155*−/− mice did not enhance *miR-155* expression in HCAEC. In support of these findings, previous studies have shown that microvesicles derived from TNF stimulated HUVEC are able to transfer *miR-155* to T lymphocytes, leading to more severe graft-vs-host disease in irradiated mice[49].

We subsequently carried out functional studies using a *miR-155*-specific antagomir and demonstrated that *miR-155* enhanced inflammatory gene expression by promoting the expression of the RELA NF-κB sub-unit. The mechanism involves BCL6, a negative regulator of NF-κB, whose expression in EC was suppressed by *miR-155* delivery. Thus, we suggest that NMVs enhance NF-κB expression in endothelial cells by delivering *miR-155*, which inhibits the negative regulator BCL6. Consistent with this, we observed that NMVs enhance NF-κB expression at atheroprone regions in hypercholesterolemic mice. Moreover, *miR-155* expression is increased in both human and mouse atherosclerotic plaques, and *ApoE*−/− mice deficient in *miR-155* have been shown to have reduced atherogenesis and monocyte recruitment[50]. Interestingly a study by Zheng et al. found that injection of exosomes derived from smooth muscle cells overexpressing *miR-155* enhanced atherosclerotic plaque formation[51], suggesting that there could be multiple sources of *miR-155* contributing to the elevation of this miRNA within plaques. We hypothesise that NMVs play a role in focal increases in *miR-155* levels leading to enhanced vascular inflammation and accelerated atherogenesis. To address this hypothesis, we injected NMVs isolated from wild type or *miR-155*−/− mice into hypercholesterolemic mice and found that only those from wild type mice were able to enhance atherosclerotic plaque formation, demonstrating that NMV delivery of *miR-155* to atheroprone regions is crucial in this response. It should be noted that NMVs contain a complex mixture of proteins, RNAs, and miRNAs that may also contribute to atherosclerosis development and future studies should address the potential role of these factors. Nevertheless, despite this complex cargo, we have shown that *miR-155* is an important component and may be a potential therapeutic target in atherosclerosis.

Together, our studies presented here provide fundamental insights into the mechanism of action of NMVs in enhancing vascular inflammation and monocyte recruitment to developing plaques, potentially solving a long-standing enigma regarding the role of neutrophils in atherosclerosis. The ability of NMVs to preferentially adhere to atheroprone sites and activate endothelial cells could be a major mechanism by which neutrophils contribute to plaque formation but are rarely detected in human plaques.

## Methods

**Ethics**. For human studies, experiments complied with all ethical regulations and were approved by the University of Sheffield Research Ethics Committee (reference SMBRER310) or the Loughborough University Ethical Subcommittee for Human Participants (Study title: Molecular and Hormonal Responses to Diet-Induced Insulin Resistance, Approval number: R13-P171). The trial was also retrospectively registered at ClinicalTrials.gov (identifier: NCT03879187). All subjects gave informed consent and the experimental procedures and possible risks were fully explained. The pre-specified primary outcomes were glucose and insulin concentrations, glucose kinetics, skeletal muscle insulin signalling and muscle microvascular eNOS content and phosphorylation. These outcomes have been published[52] and the data presented in the current manuscript is an ad-hoc analysis which was covered under the original ethical approval as all human volunteers consented to their samples being used in future research related to cardiometabolic health outcome (approved by the Loughborough University Ethical Subcommittee for Human Participants).

All procedures involving mice complied with all ethical regulations and were approved by the University of Sheffield ethics committee and performed in accordance with the UK Home Office Animals (Scientific Procedures) Act 1986 under Project Licences 40/3562 and PF8E4D623. Both male and female mice were used. Where appropriate, age-matched animals were randomly assigned to treatment groups. Number rather than treatment group was used to label samples for subsequent blinded analysis. *ApoE*−/− mice were sourced from an in-house colony derived from breeding pairs obtained from Jackson Laboratories (JAX 2052; ME). *miR-155*−/− mice were supplied by Jackson Laboratories (JAX 7745; ME).

**High fat diet**. Fifteen healthy individuals (13 males and 2 females) with a mean ± SEM age 24 ± 1 y, height 176.1 ± 2.1 cm, body mass 77.65 ± 3.02 kg and body mass index (BMI) of 24.9 ± 0.6 kg m−2 volunteered to participate in this study. Participants attended an initial pre-screening visit for assessment of baseline anthropometric characteristics and estimation of resting energy expenditure according to the calculations described by Mifflin et al.[53]. From these procedures it was determined that a daily energy intake of 13474 ± 456 kJ was required to maintain energy balance. A week long, high-fat diet intervention was carried out in order to increase

daily energy intake by ~50% (19868 ± 759 kJ). Macronutrient intake was 333 ± 14 g [64%] fat, 188 ± 8 g [16%] protein and 237 ± 8 g [20%] carbohydrate. All foods were purchased and prepared by the research team and diet adherence was assessed via daily interviews and through asking participants to return any uneaten foods. Fasting venous blood samples were obtained in the morning before commencing the high-fat diet and again after 7 days adherence. Blood samples were collected at least 12 h after consuming the previous evening meal. Platelet poor plasma was prepared by spinning platelet rich plasma at 2000 × g for 20 min, and immediately stored at −20 °C for batch analysis of MV levels at the University of Sheffield.

For mouse studies, $ApoE^{-/-}$ mice were fed chow or a high fat (21%) Western diet (8290; Special Diet Services, UK) for 6–20 weeks. Neutrophil depletion studies were carried out using i.p. anti-Ly6G antibody administration (100 μg per injection; Biologend, catalogue no: 127649)[2,54]. Differential blood counts were made using blood smears stained with a Kwik-Diff™ kit (ThermoFisher Scientific, MA). Total circulating leucocyte counts were acquired using a hemocytometer. From this the total levels of circulating neutrophils, monocytes and lymphocytes were calculated (Supplementary Fig. 11).

**NMV isolation.** NMV isolation was based on the method of Timár et al.[41] with some modifications. In brief, human neutrophils were isolated from peripheral venous blood by density gradient separation. Mouse peripheral blood neutrophils were isolated by negative immunomagnetic separation[55]. Isolated neutrophils were stimulated with the bacterial derived peptide N-formylmethionyl-leucyl-phenylalanine (fMLP 10 μmol L⁻¹; Sigma-Aldrich, MO) for 1 h (37 °C in 5% CO₂). Neutrophils were then removed by spinning twice at 500 × g for 5 min and the supernatant collected. In some experiments, PBS (containing calcium and magnesium) or acetylated LDL (acLDL 20 μg mL⁻¹; Fisher Scientific, MA) were also used as alternative stimuli. To remove residual fMLP, NMV suspensions were dialysed using dialysis cassettes (ThermoFisher Scientific, MA). To pellet NMVs, the suspension was centrifuged at 20,000 × g for 30 min. In some assays control samples were spiked with 20 μL of the supernatant from this step to assess HCAEC activation. NMV suspensions were tested for platelet contamination by flow cytometry using anti-human CD41 (10 μl per sample; BD Biosciences, UK, catalogue no: 560979) and for endotoxin contamination using Limulus Amebocyte lysate assay (Lonza, UK) and found to contain neither. NMV concentration from each isolation was quantified by flow cytometry. In order to detect NMVs, settings were standardised on an LSRII flow cytometer (BD Biosciences, UK) for forward (size) and side (granularity) scatter parameters using Megamix fluorescent calibration beads (Bio-Cytex, France) according to the manufacturer's instructions (Supplementary Fig. 12a–d). NMVs were quantified using Sphero™AccuCount beads (Saxon Europe, UK). 20 μl of sample and 10 μL of Sphero™AccuCount beads (2.04 μm; Saxon Europe, UK) with a concentration of approximately 1 × 10⁶ particles mL⁻¹, were added to 300 μl PBS. The flow cytometer was set to count 1000 beads and the number of NMV counts used to determine the concentration in the sample using the formula:

$$(A \text{ divided by } B) \times (C \text{ divided by } D) = \text{number of MVs per } \mu L$$

where:
A = number of events for sample
B = number of events for the AccuCount Particles (i.e. 1000)
C = number of AccuCount Particles per 10 μL (i.e. 10,000) and
D = volume of test sample initially used in μL (i.e. 20 μL).

For experiments where fluorescently labelled NMVs were required, PKH26 or PKH67 fluorescent cell linker kits for general cell membrane labelling were used to label the NMVs directly (Sigma, UK) according to the manufacturer's instructions. Integrity of NMV samples was checked using electron microscopy. Samples were examined (original magnification ×17000) using a Tecnai Transmission Electron Microscope (ThermoFisher Scientific, MA) at an accelerating voltage of 80 kV and micrographs were taken using a Gatan digital camera (Gatan, CA; Fig. 1a). For negative staining (Fig. 1b), 5 μL of NMV suspension was absorbed onto a glow-discharged thin-film carbon-coated copper grid for 1 min. The grid was blotted, washed with 50 μL of distilled water for 5 s, blotted and washed again. After blotting for a third time, the grid was incubated with 50 μL of uranyl formate for 20 s. The grid was blotted once more and any remaining moisture was removed using a vacuum pump. Grids were imaged immediately using a CM100 Transmission Electron Microscope (Philips, UK). Human NMV size distribution was assessed using Tunable Resistive Pulse Sensing (TRPS). Measurements were made using an iZON qNano Gold with a NP400 nanopore and SKP400 calibration beads and with the classic capture mode of the iZON Control Suite Software (version 3.2.2.268; iZON Science, New Zealand). Mouse NMV size distribution was assessed using Nanoparticle Tracking Analysis (NTA). Measurements were made using a ZetaView (Particle Metrix, Germany) with 110 nm calibration beads (Thermofisher), a frame rate of 3.75 frames s⁻¹ and shutter speed of 70. For post-acquisition analysis, parameters were set to a minimum brightness of 25 and a minimum and maximum area of 5 and 999 pixels, respectively. Measurements were taken at 11 positions in the cell, with two cycles of each position. Data was then analysed using Particle Metrix software (ZetaView 8.03.08.03). This is in accordance with the current recommendations of the International Society for Extracellular Vesicles[56] for characterisation of single vesicles. In addition, we also investigated the expression of proteins that NMVs may inherit from their origin cell, as suggested by the International Society of Extracellular

Vesicles. Methods are described in detail in the section on flow cytometry analysis of surface molecule expression.

**Preparation of mouse aortic arch homogenates.** Six-week old $ApoE^{-/-}$ mice were fed chow diet for 20 weeks, or Western diet for 6 weeks or 20 weeks. Aortic arch homogenates were prepared based on the method described by Leroyer et al.[18]. Mice were culled by i.p. injection of pentobarbital overdose and aortae perfused in situ with ice cold PBS. Aortic arches were dissected in ice cold PBS, rinsed in DMEM and minced thoroughly using fine scissors and forceps in 1 ml of DMEM. After centrifugation at 400 × g for 15 min to remove contaminating cells, the supernatant was transferred and was further centrifuged at 5000 × g for 5 min to eliminate cellular debris. The homogenate was then analysed by flow cytometry.

**Multicolour flow cytometry analysis of MVs.** MV levels in platelet poor plasma or tissue homogenates were assessed by multicolour flow cytometry to detect specific surface markers derived from the parent cell. Due to technical difficulties with antibody binding, mouse PPP samples were not analysed. MVs were detected and quantified as described above. The gating strategy for analysis of human plasma MVs is shown in Supplementary Fig. 12e and for mouse aortic arch homogenate in Supplementary Fig. 13. The number of events in the positive gate for each marker was quantified using FlowJo analysis software (Tree star Inc, Ashland, OR) and the total number of MVs in each subpopulation calculated from the total. Staining of plasma MVs and multicolour flow cytometry analysis was perfomed using the following fluorescently conjugated antibodies (all at 2 μL 100 μL⁻¹):

*Human plasma*: APC-anti- human CD41 (platelets; ThermoFisher Scientific, MA, catalogue number: 17-0419-42); PE-anti-human CD14 (human monocytes; Biologend, CA, catalogue number: 301806); PE-cyanine 7 anti-human CD144 (endothelial cells; ThermoFisher Scientific, MA, catalogue number: 25-1449-41); FITC-anti-human CD66b (neutrophils; Biologend, CA, catalogue number: 305103).
*Mouse plasma*: brilliant violet 421-anti-mouse CD41 (platelets; BD Biosciences, CA, catalogue number: 133911); PE-anti-mouse CD155 (monocytes; ThermoFisher Scientific, MA, catalogue number: 12-1550-41); APC-anti-mouse CD144 (endothelial cells; Biologend, CA, catalogue number: 138012); FITC-anti-mouse Ly6G (neutrophils; BD Biosciences, CA catalogue number: 551460).

MV content of aortic arch homogenate was assessed using the above antibodies described for mouse plasma.

Fluorescence minus one (FMO; a sample containing all of the fluorochromes apart from the one being measured) and isotype antibody control samples were run in order to set gates (to exclude background fluorescence from other fluorochromes and non-sepecifc antibody binding) for positive fluorescence in each channel. Samples were analysed using an LSRII flow cytometer (BD Biosciences, UK) with FlowJo analysis software (Tree star Inc, Ashland, OR). MVs were quantified using Sphero™AccuCount beads (Saxon Europe, UK) as described. The percentage of the population that was positive for each marker was analysed using FlowJo analysis software (Tree star Inc, Ashland, OR).

**NMV adhesion and internalisation in vivo.** To assess NMV adhesion and internalisation, saline (150 μL) or fluorescently labelled NMVs (4 × 10⁶ in 150 μL) were injected via the tail vein into $ApoE^{-/-}$ mice fed a Western diet for 6 weeks. After 2 h, mice were culled by i.p. injection of pentobarbital overdose and en face immunostaining of the mouse aortic arch was carried out. Aortic arches were flushed with PBS and perfusion-fixed with 2% formalin prior to harvesting. For BCL6 expression, carotid arteries were manually dissected, cleaned of extraneous tissue, cannulated with a micropipette-in-pipette at each end and mounted vessels incubated ex vivo with or without 4 × 10⁶ NMVs (based on the mean decrease of 3.6 × 10⁶ million MVs in neutrophil depleted mice) for 2 h. RelA expression was assessed in the aortic arches of $ApoE^{-/-}$ mice injected with 4 × 10⁶ NMVs, or an equivalent volume of saline, twice weekly via the tail vein. Fixed vessels were immunostained using primary antibodies to mouse BCL6 (2 μg mL⁻¹; Santa Cruz, TX, catalogue number: sc-7388), mouse RelA (2 μg mL⁻¹; Santa Cruz, TX, catalogue number: sc-372), or isotype control followed by Alexa 568 conjugated secondary antibodies (Life Technologies, CA). Negative control was performed with omission of secondary antibody in order to detect background fluorescence. Endothelial cells were stained with Alexa 488 or 594 anti-mouse CD31 antibody (2 μg mL⁻¹; Biologend, CA, catalogue number: 102514) and nuclei were identified with TO-PRO-3 Iodide. Stained vessels were then cut longitudinally and opened en face, mounted using Prolong gold anti-fade mountant (Life Technologies, CA) for visualisation using a confocal fluorescence microscopy (Leica TCS SP8 or Zeiss LSM510 NLO inverted microscopes). Expression was assessed by quantification of fluorescence intensity using ImageJ software (1.49 V, NIH). Z-stack image acquisition was performed with Las X software (Leica Microsystems, Germany) using a 1024 × 1024 format, with a pixel size of 80.17 nm × 80.17 nm, at 1000 Hz or 700 Hz scanning speed, and with line average of 3, zoom of 2.25 or 5× with a z-step of 0.3 μm. 3D reconstruction was performed using LasX software.

**Human monocyte isolation.** In experiments where NMVs and monocytes were both used, cells were isolated from the same donors. Following density gradient

separation, peripheral blood mononuclear cells were harvested and monocytes isolated using negative immunomagnetic separation according to the manufacturer's instructions (Monocyte Isolation Kit II, Miltenyi Biotec, Germany).

**Human coronary artery endothelial cells**. Primary human coronary artery endothelial cells (HCAEC) were obtained from Promocell (Germany). The HCAEC were isolated from the left and right coronary arteries from a single donor. Cells were cryopreserved immediately after isolation, shipped and stored in liquid nitrogen until use. Cells were then defrosted and cultured at 37 °C, 5% $CO_2$ in specialised endothelial growth cell medium (MV2, Promocell, Germany) until confluent. Primary cultures were dissociated using DetachKit (Promocell, Germany) and used for experiments at passage 4–6. Endothelial cell preconditioning was performed using the Ibidi pump system (Ibidi, Germany). HCAECs ($1.7 \times 10^5$) were seeded on to gelatin coated, 0.4 μm deep flow chambers (Ibidi μ-slide $I^{0.4}$ Leur, Ibidi, Germany) and incubated for 2 h at 37 °C and 5% $CO_2$ to allow adhesion. All materials, including slides, perfusion sets, and Ibidi units were autoclaved prior to being placed in an incubator at 37 °C and 5% $CO_2$ to equilibrate for at least 4 h prior to commencing the experiment. HCAECs were cultured under high unidirectional shear stress (HSS; 13 dyne cm$^{-2}$) or low oscillatory shear stress (OSS; 4 dyne cm$^{-2}$, oscillating at 1 Hz) for 72 h. Apoptosis and necrosis rates were measured ± incubation with NMVs ($1 \times 10^3$ μL$^{-1}$) for 2 and 4 h using a flow cytometry-based assay that measured Annexin V binding and propidium iodide uptake. The percentage of cells that were positive for fluorescence for Annexin V or propidium iodide was quantified and it was found that NMVs did not alter HCAEC apoptosis or viability, respectively (Supplementary Fig. 14a, b).

**NMV adhesion in vitro**. HCAEC were cultured under static conditions, HSS or OSS for 72 h as described above. Fresh complete growth medium containing fluorescently labelled NMVs ($1 \times 10^3$ μL$^{-1}$) was added to cells under static conditions or perfused over Ibidi μ-slides for 2 h at 37 °C and 5% $CO_2$ under the same shear conditions used for pre-conditioning the cells. Following incubation, the medium was removed and cells gently washed three times with PBS to remove residual NMVs. Phase-contrast and fluorescent images were taken using the ×20 lens of a wide-field microscope (Leica, DM14000B) and the mean number of fluorescent NMVs in six fields of view per sample was calculated.

To assess NMV adhesion to monocytes, fluorescently labelled NMVs ($1 \times 10^3$ μL$^{-1}$) were incubated with $2 \times 10^5$ monocytes for 2 h at 37 °C. Unbound NMVs were removed by centrifugation ($400 \times g$ for 6 min) and the cells washed. NMV adhesion was analysed using an LSRII flow cytometer and data analysed for changes in mean fluorescence intensity using FACSDiva acquisition software.

**Adhesion of monocytes to HCAEC under flow in vitro**. HCAEC were cultured under OSS for 72 h as described above. Unlabelled NMVs ($2 \times 10^3$ μL$^{-1}$) were perfused over the cells under OSS for 2 or 4 h. The media was removed and replaced with media containing fluorescently labelled monocytes ($1 \times 10^3$ μL$^{-1}$) and this was perfused over the HCAEC for 2 h under OSS. Slides were washed gently to remove non-adherent monocytes and fixed in paraformaldehyde (4% w/v). Phase-contrast and fluorescent images were taken using the ×10 objective of a wide-field microscope (Leica, DM14000B) to detect adherent monocytes. An average of 15 images were analysed per slide and used to calculate the number of adherent monocytes per field of view per sample.

**Monocyte transendothelial migration in vitro**. HCAEC were cultured on transwell inserts. Monolayer integrity was checked using FITC-BSA, and found to retain >80% FITC-BSA in the upper chamber of the plate after 120 min. HCAEC were incubated ± NMVs ($1 \times 10^3$ μL$^{-1}$) for 30 min. Subsequently, $2 \times 10^5$ monocytes were added to the upper chamber and the number of monocytes that had migrated into the lower chamber in response to CCL2 (5 nmol L$^{-1}$) was counted after 90 min. To determine the role of HCAEC, the experiment was repeated in the absence of HCAEC. To determine the effects of CD18 inhibition, NMVs were treated with anti-CD18 (6 μg per $10^6$ MVs; 6.5E gifted from M. Robinson, SLH Celltech Group, UK) or isotype control for 20 min at room temperature. Unbound antibody was removed by washing twice, resuspending the pellet in 1 mL of buffer followed by centrifugation at $20,000 \times g$ for 30 min, and the transendothelial migration experiment was repeated.

**Flow cytometry analysis of surface molecule expression**. NMVs were stained with anti-human FITC-conjugated CD18 (ThermoFisher Scientific, MA, catalogue number: 11-0189-42), APC-conjugated anti-human L-selectin (BD Biosciences, UK catalogue number: 561916), PE-anti-human PSGL-1 (BD Biosciences, UK catalogue number: 556055) or isotype control (all at 2 μL 100 μL$^{-1}$). For HCAEC, cells were pre-conditioned as above or cultured under static conditions. Cells were washed in PBS and trypsin (diluted 1:6 to avoid damage to ICAM-1 epitopes) was added to each slide to detach endothelial cells. After labelling with FITC-conjugated anti-ICAM-1 (5 μL 100 μL$^{-1}$; Bioloegend, CA, catalogue number: 353108) for 40 min on ice, cells were washed twice and analysed using flow cytometry. Monocytes were incubated with NMVs for 30 min, washed and stained with PE-conjugated anti-L-

selectin (2 μL 100 μL$^{-1}$; ThermoFisher Scientific, MA, catalogue number: 12-0629-42), PE-conjugated anti-CD11b (2 μL 100 μL$^{-1}$; ThermoFisher Scientific, MA, catalogue number: 12-0118-42) or isotype control for 40 min on ice and then washed twice to remove unbound antibody. FMO (where necessary) and isotype control samples were run to set gates for positive fluorescence in each channel. Mean fluorescence intensity of samples was analysed using an LSRII flow cytometer (BD Biosciences, UK) with FlowJo analysis software (Tree star Inc, Ashland, OR).

**Cytometric bead array**. HCAEC ($3 \times 10^4$) were cultured 72 h prior to the experiment in a 24-well plate. Cells were incubated ± NMVs ($1 \times 10^3$ μL$^{-1}$) and media collected at 2 h and 4 h. CCL2, IL-6 and CXCL8 levels were assessed using a cytometric bead array (BD Biosciences, UK), a flow cytometry application that allows quantification of multiple proteins simultaneously.

**Enzyme linked immunosorbent assays (ELISA)**. HCAEC were cultured under OSS for 72 h as described above. NMVs ($2 \times 10^3$ μL$^{-1}$) were perfused over the cells under OSS for 4 h. The media was removed and human CXCL8, CCL2 and IL-6 Duoset ELISA kits (R&D systems, Abingdon, UK) were used to determine the protein secretion from HCAECs cultured with NMVs.

**Western blot analysis**. HCAEC ($3 \times 10^4$) were cultured 72 h prior to the experiment in a 24-well plate. Cells were incubated ± NMVs ($1 \times 10^3$ μL$^{-1}$) for 2 h. HCAEC were then washed, lysed, and centrifuged to remove any cellular debris. Sample buffer was added, samples boiled for 5 min, run on 4–12% Bis-Tris gel (Life Technologies, Paisley UK) and transferred to nitrocellulose membranes (Millipore, UK). Membranes were blocked and exposed to mouse anti-BCL6 (1:250 dilution; Santa Cruz, CA, catalogue number: sc-7388), rabbit anti-RelA (1:1000 dilution; Santa Cruz, CA, catalogue number: sc-372), mouse anti-ICAM-1 (1:500 dilution; Bio-Techne, UK, catalogue number: MAB720-SP), goat anti-VCAM-1 (1:1000 dilution; Bio-Techne, UK, 12-1069) and mouse anti-CCL2 (1:1000 dilution; Bio-Techne, UK, catalogue number: MAB279-SP) overnight. Membranes were washed and goat anti-rabbit, goat anti-mouse or rabbit anti-goat Horseradish Peroxidase conjugated secondary antibody (Dako, UK) was added (1:2000 dilution) for 1 h. Membranes were washed again and incubated for 1 min at room temperature with ECL Prime® (GE Healthcare, UK). Membranes were imaged and the optical density analysed using a LI-COR C-DiGit® Blot scanner (LI-COR Biosciences, NE).

**RNA extraction and RT-qPCR**. Plasma MVs were isolated by centrifuging platelet poor plasma at $2000 \times g$ for 30 min and collecting pellets. NMV were prepared as described and isolated by centrifuging at $20,000 \times g$ for 30 min. HCAEC ($3 \times 10^5$) were incubated in a 6-well plate or an Ibidi μ-slide with or without MVs ($1 \times 10^3$ μL$^{-1}$) for 2 h at 37 °C. Cells were dissociated using trypsin and centrifuged at $300 \times g$ for 5 min. $ApoE^{-/-}$ mice were injected i.v. with $4 \times 10^6$ NMVs via the tail vein. After 2 h, mice were culled by i.p. injection of pentobarbital overdose and aortae perfused in situ with ice cold PBS. The aortas were dissected immediately and cut into small pieces before lysis. RNA and miRNA were extracted using RNA isolation kit (Bioline, UK) and Pure Link microRNA isolation kit (Invitrogen, CA) respectively and reverse transcription polymerase chain reaction (RT-PCR) performed with 0.1 μg of RNA or miRNA using a OneStep kit (Qiagen, Germany); quantitative PCR (qPCR) was performed using SYBRgreen (Sigma-Aldrich, MO) for RNA analysis and Taqman (Eurogentec, Belgium) for miRNA. Relative gene expression was calculated comparing the number of cycles required to produce threshold quantities of product and calculated using the ΔΔCT method. β actin was used as housekeeping gene to normalise regulation of expression.

miRNA copy number was determined by spiking samples with known amounts of custom RNA oligonucelotides (Sigma Aldrich, MO) corresponding to the mature miRNA sequences. RNA oligos were serially-diluted in RNAse free water and amplified by qPCR. A standard curve was generated from the $C_t$ value equivalent to the known amount of RNA oligo in each Taqman qPCR reaction[57,58] and the copy number was calculated using the molar mass of the synthetic RNA oligonucleotides (Supplementary Fig. 15a, b). Detail of the primers used is given in Supplementary Table 5.

**Microscopy analysis of NMV internalisation in vitro**. Live cell imaging was performed in HCAEC. Cells were seeded on glass bottomed μ-slides (Ibidi) and cultured for 24 h. CellLight® Early Endosomes-RFP or SiR-Actin (Spirochrome, Switzerland) and Hoechst (both from ThermoFisher Scientific, MA) staining was performed to detect early endosomes or F-actin and nuclei in living cells respectively. Cells were then incubated with fluorescently labelled NMVs and confocal live cell imaging performed on a Leica TCS SP8 imaging platform (Leica, Germany) equipped with an incubator, allowing live cell imaging at 37 °C and at 5% $CO_2$. Samples were observed using a ×100 oil immersion objective (HCX PL APO ×100/1.40 oil) and excited using sequential scanning, first at 405 nm and 532 or 652 nm for Hoechst and Early Endosome-RFP or SiR-actin, respectively and at 490 nm for PKH67. HyD hybrid detectors were used to detect three spectral regions: 415–496 nm (Hoechst), 588–635 nm (Early Endosomes-RFP) or 662–749 nm (SiR actin), and 500-550 nm

(PKH67). Z-stacks were performed with a z-step size of 0.20 µm. Images were analysed using ImageJ (1.49 V, NIH) and Amira 6 software (ThermoFisher Scientific, MA).

**Flow cytometry quantification of neutrophil microvesicle internalisation in vitro**. In order to quantify NMV internalisation, HCAEC ($5 \times 10^4$) were seeded onto 24-well tissue culture plates and cultured overnight. The next day HCAEC were incubated with media alone, media + TNF (1 ng mL$^{-1}$; R&D Systems, Abingdon, UK), media + anti-ICAM-1 (100 ng mL$^{-1}$; Biolegend, London, UK, catalogue number: 322704) or media + TNF + anti-ICAM-1 for 4 h. Consequently fluorescently labelled NMVs ($0.4 \times 10^3$ µL$^{-1}$) were added and cells incubated for 2 h at 4 °C, room temperature or 37 °C (as specified in the corresponding figure legends). The use of low temperatures to inhibit endocytosis has been used by others to demonstrate internalisation is a metabolically active process i.e. endocytosis dependent[59,60]. Cells were then washed to remove excess NMVs and detached using a trypsin/EDTA solution. Cells were washed and, immediately prior to flow cytometry analysis, Trypan blue (1 mg mL$^{-1}$) was added to each sample in order quench fluorescence of residual surface bound NMVs[61,62], thus ensuring any fluorescent signal detected was from internalised NMVs only. Data were analysed for mean fluorescence intensity using an LSRII flow cytometer using FACSDiva acquisition software.

**Transfection with antagomir**. HCAEC ($3 \times 10^5$) were seeded in six-well plates and, on the following day, were transfected with 25 ρmol of *miR-155* antagomir or scrambled antagomir (Creative Biogene, NY) in unsupplemented basal media for 5 h using Lipofectamine (Invitrogen, CA). Cells were then cultured for 24 h in MV2 media (Promocell, Germany) prior to the addition of NMVs, and RT-qPCR carried out as described above.

**Atherosclerotic plaque analysis**. Six-week-old *ApoE*$^{-/-}$ mice were fed a Western diet for 6 weeks. During this period, $4 \times 10^6$ mouse NMVs, or an equivalent volume of saline, were injected twice weekly via the tail vein for the entire 6 weeks or the final 2 weeks of the diet. En face staining with Oil Red O was performed and lesion coverage in aortae was analysed using NIS elements analysis software (Nikon, NY)[63]. Whole aortae were opened longitudinally and fixed and stained with Oil Red O to identify lipid-laden lesions. Areas of positive staining were selected using hue, saturation and intensity filters to determine the lesion area. Lesion assessment was blinded and areas were expressed as a percentage of the total aortic surface area. For en face staining of monocyte/macrophage accumulation, fixed vessels were immunostained using primary antibodies to mouse MAC3 (1:100 dilution; BD Biosciences, CA, catalogue number: 550292) or isotype control followed by Alexa 568 conjugated secondary antibodies (Life Technologies, CA). Negative control was performed with omission of primary antibody in order to detect background fluorescence. Endothelial cells were stained with Alexa 488 anti-mouse CD31 antibody and nuclei were identified with TO-PRO-3 Iodide. Stained vessels were then cut longitudinally and opened en face, mounted using Prolong gold anti-fade mountant (Life Technologies, CA) for visualisation using a confocal fluorescence microscopy (Leica TCS SP8 or Zeiss LSM510 NLO inverted microscopes). Expression was assessed by quantification of fluorescence intensity using ImageJ software (1.49 V, NIH).

For analysis of aortic root sections, mice were perfusion fixed (PBS with 4% PFA) after terminal anaesthesia. The upper portion of the hearts were dissected horizontally at the level of the atria and placed in 30% sucrose for 24 h before embedding in Optical Cutting Temperature (OCT) compound. Serial 7 µm sections were processed for staining with Oil Red O or for anti-Mac3 (1:100 dilution; BD Biosciences, CA, catalogue number: 550292) with Mayer's hematoxylin used as a counterstain. For analysis of monocyte/macrophage content in the aortic root, cryosections were immunostained with anti-mouse MAC3 antibody (1:100 dilution; BD Biosciences, CA, catalogue number: 550292) with DAB as the chromagen and Mayers hematoxylin as a counter stain. NIS-elements analysis software (Nikon, NY) was used to calculate the total lesion area and areas of positive staining were selected using hue, saturation and intensity.

**Statistical analysis**. Results are presented as mean ± SEM throughout. Statistical analyses were performed using GraphPad Prism version 7.00 (GraphPad Software, CA). Data was analysed assuming Gaussian distribution using paired or unpaired *t*-tests, one-way ANOVA followed by Tukey's post hoc test for multiple comparisons or Dunnett's post hoc test to compare to control values, or two-way ANOVA followed by Tukey's or Bonferonni's post hoc test for multiple comparisons. *P* values of less than 0.05 were considered significant. For in vitro experiments, *n* numbers relate to different donors for both HCAEC and neutrophils/monocytes. In experiments where percentages are shown, data are expressed as the percentage of the mean of the control samples.

**Reporting summary**. Further information on research design is available in the Nature Research Reporting Summary linked to this article.

## Data availability
The datasets generated during and/or analysed during the current study are available in the Source Data File. The source data underlying Figs. 1c, f, g–j, 2e, 3b–c, e, g–j, 4a–e, 5d–f, 6a–h, 7a, 7e, 8b, d, f, 9a, b and Supplementary Figs. 1a, b, 2a, b, 4a, b, c, 5a, b, c, 6, 7a, b, 9, 10, 11, 14a, b and 15a, b are provided as a Source data file.

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

## Acknowledgements

We thank Dr F. Stassen for the use of the qNano Gold and K. Knoops for advice using the Amira software. We thank Fiona Wright and Carl Wright for their expert technical assistance. This work was funded by: British Heart Foundation Programme Grant (CS, PE); British Heart Foundation Project Grants PG/09/067/27901 (AB, VR), PG/13/55/30365 (LW, SF), PG/14/38/30862 (CR, VR), PG/16/44/32146 (JJ, EKT, SF); British Heart Foundation Studentship FS/14/8/30605 (BW, VR); MRC Fellowship MR/K023977/1 (RB); and European Union's Horizon 2020 Marie Skłodowska-Curie Innovative Training Network, TRAIN 721532 (CN).

## Authors contribution

I.G., B.W., C.S., M.A. and C.R. contributed to the design of experiments, acquisition and analysis of data and preparation of the paper. J.J., A.B., M.M., L.A.L., L.W., M.L., S.P., R. W., C.H., C.N. and M.v.Z. contributed to the acquisition and analysis of data. L.W., B.B., R.B., S.F., E.K.-T. and A.S. contributed to the acquisition of data. P.H. contributed to the initial design of the study and reviewed the paper. P.C.E. contributed to the conception and design of experiments, the analysis of data, the preparation of the paper. V.R. conceived the study, designed the experiments, contributed to the acquisition of data, analysed the data and wrote the paper. All authors reviewed the paper and provided intellectual content.

## Competing interests

The authors declare no competing interests.
