## [Peer Review File · Nature Communications]

Reviewers' Comments:

Reviewer #1:

Remarks to the Author:

In this study entitled "Neutrophil microvesicles drive atherosclerosis by delivering miR-155 to atheroprone endothelium", Gomez and colleagues assess the potential contribution of neutrophil derived miR-155 in regulating EC activation during atherosclerosis. The authors suggest that neutrophil microvesicles promote EC inflammation by delivering miR-155, which has been shown to regulate EC activation. Overall the manuscript do not present solid data to support author's main conclusions. Moreover, the atherosclerosis analysis is very poor, based in a limited number of mice (Figure 7; n=7). All atherosclerosis research experts suggest at least 12 mice per group to raise a solid conclusion. The authors only present en face analysis of neutral lipid deposition (ORO staining). This analysis is preliminar and only inform about neutral lipid accumulation in the whole artery. Additional analysis (aortic root and brachiocephalic artery crosssection analysis) is needed to support author's conclusions. The authors should also assess macrophage content in early stage lesions. If miR-155 derived from neutrophils regulates EC activation, mice treated with NMV will present more macrophages content in atherosclerotic lesions. In addition to these key experiments, the authors should also address the following concerns:

- 1) Figure 3B. The conclusion of the authors is based in a limited number of experiments. How ICAM expression was calculated?.
- 2) Figure 3E. The result is based in limited number of experiments, which have a significant variability. The attachment of monocyte should be normalized by the number of EC attached to the plate. The authors should repeat this experiment labelling previously the ECs (celltracker) and incubating the cells with labelled monocytes. Then, the authors should normalize the number of monocytes per EC.
- 3) miR-155 is highly expressed in activated monocytes, suggesting that the specific contribution of miR-155 derived from neutrophil could be minimal.
- 4) Figure 6B. The authors should demonstrate that the uptake of miR-155 via NMV internalization is incorporated to the Ago complex and target specific mRNA targets. To this end, the authors need to isolate neutrophil microvesicles from WT and miR-155 deficient mice. Then, treat ECs with different neutrophil microvesicles and pulldown the Ago complex. Finally assess the relative enrichment of miR-155 and miR-155 target mRNAs.
- 5) Figure 7A. To directly demonstrate the relevance of NMV derived miR-155 during the progression of atherosclerosis. The authors should isolate NMV from WT and miR-155 KO mice (available at Jackson Labs) and perform a similar series of experiments.
- 6) Figure 7C. As noted above, the authors need to perform numerous experiments to support the main conclusion of the paper

Reviewer #2:

Remarks to the Author:

The study by Gomez et al is a nice work that unraveled a yet not understand question : How microvesicles from neutrophil can be found in the wall of the atherosclerotic lesion while there is very few neutrophils found inside the lesion? They additionally provided answers on their role in the progression of the disease. They identified miR155 as a miR carried by the NMVs and transfer to the endothelial cells to regulate their inflammatory potential through NFkB regulation. Interestingly they identify that a « high fat diet » in human increases the circulating level of these MVs showing a direct evidence of the influence of modern life style on microvesicles production.

Overall, the manuscript is robust. The MVs identification and characterization that leave no doubt on what they identify. Just a comment can be made on the choice of their control condition « saline » (see major comments). Additionally their experiments were well designed, especially the way they assess copy number for miR in MVs. The imaging of the interaction with fluorescent labeling of the MVs is a very interesting/novel observation as this is usually tricky to observe. Additionally, the authors nicely integrated the flow component in their in vitro work, this gives a high value to their observation as it improves in vitro experiment to have conditions closer to in vivo condition. Nevertheless there is a doubt on if the MVs interaction with cells has been done under flow or on flow exposed cell after flow exposure. This point would be essential to clarify and answer to strengthen the manuscript. As well, the inflammatory profile upon NMVs interaction could be done under flow.

Despite these interesting findings, the current manuscript lacks novelty and implication for a broad community.

The role of miR155 in the endothelium is already known, as well as its role on NFkB. (J Cell Biochem. 2014 Nov;115(11):1928-36. doi: 10.1002/jcb.24864. / Mol Med. 2017; 23: 24-33. / Arterioscler Thromb Vasc Biol. 2013 Mar;33(3):449-54. doi: 10.1161/ATVBAHA.112.300279. Epub 2013 Jan 16.)

The ability of MVs to transfer content to endothelial cell through ICAM-1 too (both for NMVs and other MVs) as well as the expression of CD18 by NMVs. (J Am Soc Nephrol. 2012 Jan;23(1):49-62. doi: 10.1681/ASN.2011030298. Epub 2011 Nov 3.)

Moreover miR155 has been often found in circulating vesicles. (Oncotarget. 2017 Apr 4;8(14):23360-23375. doi: 10.18632/oncotarget.15579.)

Its transfer or increase in endothelial cells is not new even if it is the first time that NMVs are shown to transfer it. (Mol Ther. 2017 Jun 7;25(6):1279-1294. doi: 10.1016/j.ymthe.2017.03.031.)

The authors should also discuss more the significance and impact of their work and discovery.

Major comments

Page 5, Line 36

The author generated in vitro NMVs by stimulation of neutrophils with a bacterial compound. Why are they using a bacterial compound and not a lipid treatment as the diet in vivo ? The author could stimulate neutrophils with LDL and verify if it gives the same increase in MV production and MV content in terms of miR and CD18 expression. This would increase the relevance of the in vitro analysis.

Figure 1

The authors nicely showed that a fat diet increases MVs concentration in mice, they also showed that this amount of MVs is reduced in mice lacking neutrophils, this highly suggests that the loss of MVs observed is due to loss of NMVs but does not prove it.

The author should quantify the circulating NMVs directly and compare it between control and western diet mice (or at least state the limitation and speculation in the manuscript).

This is linked with 3 other questions :

How did the author validate neutrophils depletion?

Did the author check that neutrophils deletion is not affecting other immune cell number or the release of MVs by other cell types?

Which percentage represents the NMVs compared to other cell type MVs in the whole plasma MVs content (human and mouse)?

Figure 3

It is unclear if the adhesion was done under flow. If yes, the authors should clarify and also emphasize it in the results as this is very relevant for in vivo comparison.

If not done under flow, it would be good to assess the role of flow on direct endothelial cell-MVs contact and adhesion and internalization.

Supplemental movie + Page 8, Line 5

I don't see the importance of the live imaging to assess internalisation. I would highly recommend a co-staining with endocytosis molecules. Additionally the resolution of these videos is poor and it looks like the MVs could be localized below the cell on the basal side rather than internalized.

Figure 2, Page 9, Line 26

The author injected a certain amount of NMVs to raised the in vivo level, how much this increase the NMVs content in the circulation? Does it correlates with the increase observed at 20weeks in mice or after a week in human ?

The authors mentioned later in the manuscript that the injection increased by 30% the level of circulating MVs as observed in human (page 11, line 35). If this is already valide here, it should be stated here and even emphasized to increase the significance of the work.

Figure 4

This figure is very nice and interesting. nevertheless, for increased significance of the work, these experiment should be done on low/oscillatory flow conditioned cells as these parameters can be already modulated by flow itself.

Figure 5, Page 10, Line 52 :

I have a naive comment : I need more explanation on why decreasing the temperature allow the authors to state about an active process, I understand that lowering the temperature will reduce biological activity in the sample an should favor passive event but decreasing the temperature will also change the passive interaction especially lipids interaction so I am not sure about the strict conclusion of the author, I would suggest to nuance it. Maybe a comment in the methods would be useful for the justification of the experiment.

When the author tested the role of ICAM-1, they assumed that TNF α increases ICAM-1 expression (which is very fare in respect with the literature), nevertheless one simple additional control would have erased any doubt about the increased internalisation upon TNF α stimulation due to an other pathway : combining TNF α treatment with ICAM-1 blocking antibody. The author should add this condition to fully conclude on the specific role of ICAM-1.

Page 12, Line 37

The author talked about accumulation in the wall in their discussion but showed internalization in ECs, Is this internalisation do not lead to degradation? How then do you explain accumulation in the wall?

The Human analysis is a powerful point of the manuscript, however the author should discuss the difference between the short term diet and long term diet in mice.

The author always use a saline solution as a control of their MVs preparation. I would highly suggest to use supernatant of the MVs pellet. Especially for the experiment assessing the adhesion of NMVs to ensure that no other elements contained in the plasma but not MVs can cause the observed dot for the adhesion. Did the author check that the labelling agent is not forming cristals that could be confused with MVs ?

Minor comments

Introduction

Line 25 : « shear » alone is not proper English, please use « shear stress »

Line 26 : the reference to regulation of inflammation by Nf κ B should be moderated, not only Nf κ B control inflammation in EC.

Methods

Page 5

Line 3: Mouse : could the authors mentioned if they use males and females or only males and justify their choice?

Line 47 : it is unclear how the MVs were labelled, was it the cells which were labelled or the MVs directly ? Please clarify.

Page 6

Line 3: mention which supplements.

Line 5: when does the western diet star? (which age for the mice)

Page 7

Line 22: Can the authors specify how the washings of the MVs are done. Centrifugation ?

Page 8

Line 40 : Usually results for animal data are shown as median +IQ not mean + SEM. Can the authors refer to the recommendation of NatComm and adjust if necessary?

Can the author also stat that they assumed gaussian distribution every time they used a T-Test in the figure legend or correct the statistics by using a test without assumption of a gaussian distribution ?

Results

Page 9

Line 4: I would mentioned the characterization of MVs in the results part (figure 1A-B-C), this would avoid starting with a figure referring to methods and would reinforce the results part.

Line 9 : « curculating » is written instead of circulating

Line 23 : The author mentioned neutrophils, platelets and monocytes MVs. They also analyzed the endothelial one but do not comment about this population. This should be included.

Line 50 : figure 3D needs representative images

Page 10

Line 36 : please show the data about cytokine content as supplemental

Page 11

Line 27 : Are the authors sure they mean « deduce » ?

Line 31 : the quality of the images is not good enough, please split channels to see the BCL6 channel alone (black and with)

Line 46 : there is a missing word in the sentence

Discussion

Page 12

Lines 23-29 : this paragraph is not clear and I don't understand how this is useful for the discussion of the results. Please clarify.

Figure. 1E. We don't know if this analysis was done at 6w of diet or 20w.

Expanded methods

Multicolor flow cytometry :

After CD144, the authors mentioned « platelet » while this is an endothelial marker.

Reviewer #3:

Remarks to the Author:

Rodger et al report that Western diet can stimulate the production of circulating micro vesicles in both humans and mice. The microvesicles, particularly those derived from neutrophils (NMV), have an atherogenic effect both in vivo and in vitro. The non-coding small RNA miR155 is one of the cargos further characterized in the neutrophil derived microvesicles. This small non-coding RNA apparently is delivered to endothelial cells, enhances NF-kappaB activation, promotes monocyte adhesion and translocation, possible contributing to plaque formation and inflammation. Overall, the manuscript is well written and the experiments are well designed and comprehensive.

Major concerns:

There is a lack of some important controls. Despite the dialysis to clear the NMV remnants of fMLP, this molecule can have important effects on endothelial cells; for instance, 2,000-fold lower concentration of fMLP than the one used in this work, has been described to induce HUVEC proliferation (Langeeggen et al 2001. *Inflammation* (25):83-89). This could play a role in explaining the absence of effect from NMVs prepared from non-stimulated neutrophils.

Are there significantly lower levels of miR155 in the non-stimulated NMVs (I could not find miR155 level comparisons for Unstimulated and stimulated vesicles).

How effective are NMVs once mixed with the other MVs, mimicking more effectively what happens in vivo?

Is the plasma MV from the individuals exposed to high fat able to inhibit in vitro BCL6 or stimulate RELA in HCAECs?

It would be very relevant to quantify if these circulating MVs are present in individuals at risk for atherosclerosis development compared with matched healthy controls.

Can the authors at least speculate on what component of the diet, without adding fMLP, would induce NMV production? This is important because unstimulated NMVs do not have proatherogenic effects. Would more physiologic stimuli (Mitochondrial-derived fMLP, oxidized lipoproteins, etc.) have similar effects?

Minor concerns:

1. A picture of the murine NMV with their size characterization is missing.

2. Stats in which a t-test is used require normal distribution of the data.

3. In Fig.2 it is difficult to see the bars sizes.

4. Fig.3 resolution of fluorescence pictures is not good some of them are pixelated.

5. Fig.5 D the green background in the image is interfering with proper visualization of NMVs.

Fig.6A are these small RNAs the only ones detected or are they the only ones available for the analysis?

Fig.6B the difference in miR155 is not particularly high, are the pre-diet MVs able to activate the endothelial cells?

Fig.5.C and D should those levels of miR155 be in D similar or less than in MV?

In the supplementary table, s indicate the cell associated with the marker measure and show the averages for each column.

Gomez et al rebuttal letter

We were delighted to note that the editor and reviewers found our work “robust” and “well designed” and that it contains “interesting/novel observations” and would be willing to consider a revised version. We are grateful to the reviewers for their particularly thorough appraisal of our work. We have addressed all of the points raised through additional experimentation. This work has generated a substantial amount of additional data resulting in the insertion of six new figures and ten additional data panels. These additional data and other revisions add to the robustness of our data, highlight the novelty of the study and strengthen our conclusions.

Key observations included in the revision are:

1. *miR-155*^{-/-} neutrophil-derived microvesicles (NMVs) do not enhance atherosclerotic plaque formation or macrophage recruitment in hypercholesterolemic mice, whereas those derived from wild type neutrophils do, demonstrating the role of *miR-155*
2. neutrophils stimulated with fMLP or acLDL release larger numbers of MVs with higher *miR-155* content than unstimulated cells
3. supernatants from NMV pellets are unable to induce activation of human coronary artery endothelial cells (HCAEC)
4. NMVs induce endothelial cell activation under oscillatory shear stress conditions (OSS) and shear stress influences the internalisation of NMVs with greater numbers internalised under OSS compared to static or high shear stress (HSS)

Having addressed each of the Reviewer’s concerns, we feel that the manuscript is now much stronger and we hope it is now suitable for publication in Nature Communications. A marked-up version of the manuscript is attached where changes appear in red. Please see below for a point-by-point response to each of the Reviewer’s comments.

Reviewer #1:

In this study entitled “Neutrophil microvesicles drive atherosclerosis by delivering miR-155 to atheroprone endothelium”, Gomez and colleagues assess the potential contribution of neutrophil derived miR-155 in regulating EC activation during atherosclerosis. The authors suggest that neutrophil microvesicles promote EC inflammation by delivering miR-155, which has been shown to regulate EC activation. Overall the manuscript do not present solid data to support author’s main conclusions. Moreover, the atherosclerosis analysis is very poor, based in a limited number of mice (Figure 7; n=7). All atherosclerosis research experts suggest at least 12 mice per group to raise a solid conclusion.

We thank the Reviewer for their comments and suggestions. We have added additional *in vivo* analysis of atherosclerosis to the manuscript using an increased number of animals (n=10, as agreed on discussion with the editor). We also agree that all animal experiments should be designed robustly, using statistical models for power calculation and results reported accordingly. The new *in vivo* data is presented in Figure 9.

The authors only present en face analysis of neutral lipid deposition (ORO staining). This analysis is preliminar and only inform about neutral lipid accumulation in the whole artery. Additional analysis (aortic root and brachiocephalic artery crosssection analysis) is needed to support author’s conclusions.

We have now performed these additional experiments using NMVs isolated from wild type and *miR-155*^{-/-} mice and have analysed aortic root plaques by quantifying lipid content (Oil Red O) and macrophages (MAC3). These data definitively show that *miR-155* cargo in NMVs is pro-atherogenic in the aortic root (new data, Figure 9). We have also analysed cross sections of the brachiocephalic artery but did these did not contain detectable plaques after 6 weeks on Western diet.

[Redacted]

The authors should also assess macrophage content in early stage lesions. If miR-155 derived from neutrophils regulates EC activation, mice treated with NMV will present more macrophages content in atherosclerotic lesions.

We have now carried out MAC3 staining of aortic root sections taken from the mice injected for 2 weeks with WT or *miR-155*^{-/-} NMVs in our atherosclerosis model. Plaques of mice treated with WT NMV do indeed present with more macrophage content than *miR-155*^{-/-} NMVs as suggested by the Reviewer (Figure 9).

In short, these new data provided further evidence for our main conclusion that NMVs transfer bioactive *miR-155* into the endothelium of the developing atheroma and that this cargo is pro-atherogenic. We thank the Reviewer for suggesting these new experiments as they enabled us to improve our manuscript overall.

In addition to these key experiments, the authors should also address the following concerns:

1) Figure 3B. The conclusion of the authors is based in a limited number of experiments. How ICAM expression was calculated?.

For our experiments we used both HCAEC and NMVs from different donors and therefore required a new HCAEC donor for each repeat and were therefore reliant on the availability of suitable HCAEC donors from our commercial source. We chose this approach rather than using HCAEC from the same donor at different passages as we

believe this enables us to make more robust and physiologically relevant conclusions. ICAM-1 expression was analysed by flow cytometry using a fluorescently labeled anti-ICAM-1 antibody as described in the Methods in the Supplemental Material (Page 4 line 17). The data in Figure 3C shows the mean fluorescence intensity (MFI) of each sample as determined by FlowJo analysis software. We have added some text to the legend of Figure 3 (Page 25 line 9) to clarify this.

2) Figure 3E. The result is based in limited number of experiments, which have a significant variability. The attachment of monocyte should be normalized by the number of EC attached to the plate. The authors should repeat this experiment labelling previously the ECs (celltracker) and incubating the cells with labelled monocytes. Then, the authors should normalize the number of monocytes per EC.

Thank you for the suggestion. We have now reanalyzed the images shown in Figure 3f and have normalized to the number of HCAECs in each analysis field of view. Quantification is presented in Figure 3g. The conclusion from the experiment is not changed by this new analysis.

3) miR-155 is highly expressed in activated monocytes, suggesting that the specific contribution of miR-155 derived from neutrophil could be minimal.

We respectfully disagree that the contribution of the *miR-155* derived from the neutrophil is minimal. Following this Reviewer's suggestion, we now show that *miR-155* contained in NMVs is able to enhance plaque development (Figure 9). Additionally, NMVs increase monocyte recruitment (Figure 3g-h, Figure 8e-f and Figure 9b) and therefore this may well augment the *miR-155* content in the plaque as these cells do indeed express high levels of *miR-155*.

4) Figure 6B. The authors should demonstrate that the uptake of miR-155 via NMV internalization is incorporated to the Ago complex and target specific mRNA targets. To this end, the authors need to isolate neutrophil microvesicles from WT and miR-155 deficient mice. Then, treat ECs with different neutrophil microvesicles and pulldown the Ago complex. Finally assess the relative enrichment of miR-155 and miR-155 target mRNAs.

We agree that determining whether *miR-155* is incorporated into the Ago complex would add to our observations that *miR-155* is enriched in HCAECs when exposed to NMV. We have attempted to address this comment by using a commercially available kit (Wako microRNA isolation kit, Human Ago2) to analyse lysates from HCAEC incubated with either wild type or *miR-155*^{-/-} mouse NMVs for 2 h. Whilst we were able to detect the housekeeping gene, U6 small nuclear 1 (a non-coding RNA widely used as housekeeping in qPCR for miRNAs), we were unable to detect any *miR-155* in any of our samples and therefore cannot definitively demonstrate that *miR-155* is incorporated into the Ago complex in HCAEC. We speculate that this may be due to the relative insensitivity of the assay. However, our antagomiR data (Figure 6h) suggests that *miR-155* is indeed responsible for the alterations in gene expression.

5) Figure 7A. To directly demonstrate the relevance of NMV derived miR-155 during the progression of atherosclerosis. The authors should isolate NMV from WT and miR-155 KO mice (available at Jackson Labs) and perform a similar series of experiments.

Thank you for the suggestion. We have performed these extensive experiments and the data are shown in Figure 9. We consulted the Editor and Reviewers to refine the study design in order to optimise animal numbers whilst retaining statistical power for our analysis. 20 x 6 week old *ApoE*^{-/-} mice were fed a Western diet for 4 weeks to initiate plaque progression. The mice were then fed Western diet for a further 2 weeks in conjunction with twice weekly i.v. injection of NMVs isolated from wild type or *miR-155*^{-/-} mice (this detail is added to the Methods section on Page 9 line 1). We then analysed frozen sections of the aortic root for Oil Red O and MAC-3 staining to evaluate plaque

area and macrophage content and concluded that miR155 containing NMVs accelerated atherogenesis (Figure 9).

6) Figure 7C. As noted above, the authors need to perform numerous experiments to support the main conclusion of the paper

Please see response above regarding animal numbers.

Reviewer #2:

The study by Gomez et al is a nice work that unraveled a yet not understand question : How microvesicles from neutrophil can be found in the wall of the atherosclerotic lesion while there is very few neutrophils found inside the lesion? They additionally provided answers on their role in the progression of the disease. They identified miR155 as a miR carried by the NMVs and transfer to the endothelial cells to regulate their inflammatory potential through NFkB regulation.

Interestingly they identify that a « high fat diet » in human increases the circulating level of these MVs showing a direct evidence of the influence of modern life style on microvesicles production.

Overall, the manuscript is robust. The MVs identification and characterization that leave no doubt on what they identify. Just a comment can be made on the choice of their control condition « saline » (see major comments). Additionally their experiments were well design, especially the way they assess copy number for miR in MVs. The imaging of the interaction with fluorescent labeling of the MVs is a very interesting/novel observation as this is usually tricky to observe. Additionally, the authors nicely integrated the flow component in their in vitro work, this gives a high value to their observation as it improve in vitro experiment to have conditions closer to in vivo condition. Nevertheless there is a doubt on if the MVs interaction with cells has been done under flow or on flow exposed cell after flow exposure. This point would be essential to clarify and answer to strengthen the manuscript.

As well, the inflammatory profile upon NMVs interaction could be done under flow.

Despite these interesting findings, the current manuscript lacks novelty and implication for a broad community.

The role of miR155 in the endothelium is already known, as well as its role on NFkB. (J Cell Biochem. 2014 Nov;115(11):1928- 36. doi: 10.1002/jcb.24864. / Mol Med. 2017; 23: 24–33. / Arterioscler Thromb Vasc Biol. 2013 Mar;33(3):449-54. doi: 10.1161/ATVBAHA.112.300279. Epub 2013 Jan 16.)

The ability of MVs to transfer content to endothelial cell through ICAM-1 too (both for NMVs and other MVs) as well as the expression of CD18 by NMVs. (J Am Soc Nephrol. 2012 Jan;23(1):49-62. doi: 10.1681/ASN.2011030298. Epub 2011 Nov 3.) Moreover miR155 has been often found in circulating vesicles. (Oncotarget. 2017 Apr 4;8(14):23360-23375. doi: 10.18632/oncotarget.15579.)

Its transfer or increase in endothelial cells is not new even if it is the first time that NMVs are shown to transfer it. (Mol Ther. 2017 Jun 7;25(6):1279-1294.)

The authors should also discuss more the significance and impact of their work and discovery.

We agree with the Reviewer that these studies support our findings and we have added more to the discussion to highlight this (Page 15 line 44 and Page 16 lines 18 and 31). The novelty of our work is that we show for the first time that NMVs are able to preferentially adhere to endothelial cells in areas of disturbed flow, increasing the level of *miR-155* and leading to increased vascular inflammation. Whilst the above literature does indeed support aspects of our study, none of these works addressed the role of NMVs in atherosclerosis. Hong et al (J Am Soc Nephrol. 2012) show that under static conditions that NMVs can bind to HUVEC using CD18, increasing ICAM-1 and IL-6 expression but we have investigated this under physiologically relevant flow conditions present in atherosclerosis using primary aortic cells and have further investigated the mechanism. Zheng et al (Mol Ther. 2017) investigated the effect on atherosclerotic plaque development of injecting exosomes derived from smooth muscle cells that were transfected to overexpress *miR-155*. Zhang et al (Oncotarget. 2017) found that HUVEC derived MVs transfer *miR-155* to T cells and exacerbate graft-vs-host disease. Whilst

these data are consistent with our findings, the context was different and we believe that they do not negate the novelty of our study. As the Reviewer points out, we explain the potential role of neutrophils in atherosclerosis, unraveling why they are rarely detected in the plaque but still capable of playing a role in plaque development.

Major comments Page 5, Line 36

The author generated in vitro NMVs by stimulation of neutrophils with a bacterial compound. Why are they using a bacterial compound and not a lipid treatment as the diet in vivo? The author could stimulate neutrophils with LDL and verify if it give the same increase in MV production and MV content in term of miR and CD18 expression. This would increase the relevance of the in vitro analysis.

We thank the Reviewer for this helpful comment and agree that investigating a stimulus more relevant to diet would be useful. We used fMLP as this is relevant to infection and has been used in multiple studies investigating NMVs.

We addressed this point by investigating the numbers and content of NMVs released by resting neutrophils (PBS) and those stimulated with fMLP and acLDL and have shown that, although all NMVs express CD18, higher numbers of NMVs with relatively more *miR-155* content are released from neutrophils stimulated with either fMLP or acLDL (presented in Supplemental Figures 8b, 14 and 13 respectively). We have also added this detail to the Methods (Page 5, line 37) and Results (Page 11 line 10 and Page 12 line 37).

Figure 1

The authors nicely showed that a fat diet increase MVs concentration in mice, they also showed that this amount of MVs is reduced in mice lacking neutrophils, this highly suggests that the loss of MVs observed is due to loss of NMVs but do not prove it. The author should quantified the circulating NMVs directly and compared it between control and western diet mice (or at least state the limitation and speculation in the manuscript).

For unknown technical reasons we were unable to directly label mouse plasma MVs with the neutrophil specific marker Ly6G. We therefore used an alternative approach and depleted neutrophils from the mouse circulation and showed that there was an approximately 30% decrease in the level of circulating MVs in mice on Western diet (Figure 1i). We have added this explanation to the Methods (Page 6 line 21) and Results (Page 10, line 23).

This is linked with 3 other questions :

How did the author validate neutrophils depletion?

We performed total and differential cell counts on samples of blood taken from mice treated with anti-Ly6G to ensure neutrophils were depleted effectively without affecting other immune cells. This data is now presented in the new Supplemental Figure 1 and this detail added to the Methods section (Page 5 line 28).

Did the author check that neutrophils deletion is not affecting other immun cell number or the release of MVs by other cell types?

Yes, we did test this and now present the data in Supplemental Figure 1; see response above.

Which percentage represent the NMVs compared to other cell type MVs in the whole plasmatic MVs content (human and mouse)?

We have added pie charts to Supplemental Figures 5 and 6 and added text to the Results section (Page 10 lines 20 and 37) to demonstrate the relative percentages of circulating MVs and those present in the mouse aorta in the hope that this improves the presentation of our results. As explained above, we were unable to label mouse plasma NMVs due to technical reasons but the data in Figure 1i shows that we obtained a 30% reduction in circulating MVs in neutrophil depleted mice.

Figure 3

It is unclear if the adhesion was done under flow. If yes, the authors should clarify and also emphasize it in the results as this is very relevant for *in vivo* comparison.

If not done under flow, it would be good to assess the role of flow on direct endothelial cell-MVs contact and adhesion and internalization.

Thank you for the query. This is very important. The adhesion experiments were performed under flow for added relevance to *in vivo* settings. We have now made this clearer in the Methods (Page 6 lines 48, 50 and 51), Results (Page 11 line 5) and Figure legend to Figure 3. We have also added new data showing the effect of shear stress on NMV internalization by HCAEC (Figure 5f).

Supplemental movie + Page 8, Line 5

I don't see the importance of the live imaging to assess internalisation. I would highly recommend a co-staining with endocytosis molecules. Additionally the resolution of these videos is poor and it looks like the MVs could be localized below the cell on the basal side rather than internalized.

We used live cell imaging as we were unsure of the kinetics of NMV internalization and wanted to track the movement of NMVs over time. However, we agree that the movies were low resolution and have removed them and that the images in Figure 5a and b were difficult to interpret and have replaced them with images of cells labelled with an early endosome marker (CellLight® Early Endosomes-RFP). We see NMVs distributed throughout HCAECs and some associated with the endosomal marker. We have added this detail to the Results section (Page 12 line 23).

Figure 2, Page 9, Line 26

The author injected a certain amount of NMVs to raised the *in vivo* level, how much this increase the NMVs content in the circulation? Does it correlates with the increase observed at 20 weeks in mice or after a week in human ?

The authors mentioned later in the manuscript that the injection increased by 30% the level of circulating MVs as observed in human (page 11, line 35). If this is already valide here, it should be stated here and even emphasized to increase the significance of the work.

We thank the Reviewer for raising this point. This was an oversight and we have now emphasised that injection of NMVs was set to correlate with the 30% increase in humans at this point (Page 10 line 43) as well as later (Page 13 line 23).

Figure 4

This figure is very nice and interesting. nevertheless, for increased significance of the work, these experiment should be done on low/oscillatory flow conditioned cells as these parameters can be already modulated by flow itself.

We thank the Reviewer and agree with that investigating changes under flow would increase the significance of the work. We have now included this additional data into revised Figure 4 b, d and e and added a section on ELISA to the Methods (Page 7 line 40). We found that adding NMVs to the perfusion media increased both protein and RNA levels of inflammatory markers. We have added this detail to the Results section (Page 11 line 53, Page 12 line 6 and line 18).

Figure 5, Page 10, Line 52 :

I have a naive comment : I need more explanation on why decreasing the temperature allow the authors to state about an active process, I understand that lowering the temperature will reduce biological activity in the sample and should favor passive event but decreasing the temperature will also change the passive interaction especially lipids interaction so I am not sure about the strict conclusion of the author, I would suggest to nuance it. Maybe a comment in the methods would be useful for the justification of the experiment.

The use of low temperatures to inhibit endocytosis has been used by others to demonstrate that it is a metabolically active process (Kawamoto et al., 2012, PLoS One, 7, e34045; Schneider et al., 2017, J Biol Chem, 292, 20897-20910). Therefore by carrying out our experiments at 4 °C we could determine whether NMV internalisation was a metabolically active process i.e. endocytosis dependent. We have added this information to the Methods (Page 8 line 38) and Results sections (Page 12, line 27).

When the author tested the role of ICAM-1, they assumed that TNF α increases ICAM-1 expression (which is very rare in respect with the literature), nevertheless one simple additional control would have erased any doubt about the increased internalisation upon TNF α stimulation due to an other pathway : combining TNF α treatment with ICAM-1 blocking antibody. The author should add this condition to fully conclude on the specific role of ICAM-1.

Thank you for the comment. We have added this additional data to the Figure 5e and show that adding the ICAM-1 blocking antibody to TNF treated cells reduces NVM internalization, as postulated by the Reviewer. We have also added additional information to the Methods section (Page 8, line 34).

Page 12, Line 37

The author talked about accumulation in the wall in their discussion but showed internalization in ECs, Is this internalisation do not lead to degradation? How then do you explain accumulation in the wall?

We speculate that with increased inflammation in the vessel wall there is increased adhesion and internalization of NMVs leading to an increase in the number of NMVs present (Page 15 paragraph beginning at line 41). However, little is known about the degradation of NMVs and it is possible that this may not be a rapid process and therefore NMVs accumulate. We have added a comment relating to this in the discussion (Page 14 line 30) and have altered the text in the Results section (Page 10, line 34 and line 39) and figure legend for Figure 1 so that it does not refer to accumulation but rather the number of NMVs in the vessel wall.

The Human analysis is a powerful point of the manuscript, however the author should discuss the difference between the short term diet and long term diet in mice.

We agree with the Reviewer that this is an important point and we have added some discussion to the manuscript (Page 15 line 24).

The author always use a saline solution as a control of their MVs preparation. I would highly suggest to use supernatant of the MVs pellet. Especially for the experiment assessing the adhesion of NMVs to ensure that no other elements contained in the plasma but not MVs can cause the observed dot for the adhesion. Did the author check that the labelling agent is not forming crystals that could be confused with MVs ?

In the *in vivo* adhesion experiments we compared the atheroprotected and atheroprone regions of the aortic arch within the same animals that had been injected with fluorescent NMVs. We have added some text to the Results section (Page 10 line 45) to emphasise this. We did check that the labeling agent was not forming crystals in the *in vitro* experiments (now in Supplemental Figure 12) and have made this clearer in the legend. We have now also included images of atheroprone and atheroprotected regions of aorta after i.v. injection with supernatant containing excess dye. We did not detect any fluorescence in the PKH channel in these samples. These images are included in new Supplemental Figure 7 and we have added text to the Results section (Page 10 line 49).

Minor comments Introduction

Line 25 : « shear » alone is not proper English, please use « shear stress »

Thank you, this has been corrected (Page 4 line 24).

Line 26 : the reference to regulation of inflammation by NfκB should be moderated, not only NfκB control inflammation in EC.

We have now moderated this sentence (Page 4 line 26).

Methods

Page 5

Line 3: Mouse : could the authors mentioned if they use males and females or only males and justify their choice?

We used both male and female mice. This information has now been included for clarity in the method section (Page 5 line 10). Thank you for highlighting this omission.

Line 47 : it is unclear how the MVs were labelled, was it the cells which were labelled or the MVs directly ? Please clarify.

We have added some text to the Methods section (Page 5, line 52) to clarify that NMVs were labeled directly.

Page 6

Line 3: mention which supplements.

We have added this information to this section (Page 6, line 7)

Line 5: when does the western diet start? (which age for the mice)

We start the diet at 6 weeks old. This detail has been added (Page 6, line 10).

Page 7

Line 22: Can the authors specify how the washings of the MVs are done. Centrifugation ?

NMVs are washed by addition of buffer followed by centrifugation. This detail has now been added to the appropriate Methods section (Page 7 line 27).

Page 8

Line 40 : Usually results for animal data are shown as median +IQ not mean + SEM. Can the authors refer to the recommendation of NatComm and adjust if necessary?

Nature Communications guidelines do not specify how animal data should be presented. Fang et al {Fan et al., 2019, Nat Commun, 10, 425}, for instance, have recently published in Nature Communications investigating plaque size in *ApoE*^{-/-} mice and presented their data as mean ± SEM. However, should the Editor's and the Reviewer feel that it is more appropriate to present our results as median ±IQ, we can modify our figures accordingly.

Can the author also stat that they assumed gaussian distribution every time they used a T-Test in the figure legend or correct the statistics by using a test without assumption of a gaussian distribution ?

We have added this to the figure legends and Statistical Analysis section in the Methods (Page 9 line 13).

Results Page 9

Line 4: I would mentioned the characterization of MVs in the results part (figure 1A-B-C), this would avoid starting with a figure

referring to methods and would reinforce the results part.

We thank the Reviewer for this useful suggestion and have now included this in the Results section (Page 10 line 2).

Line 9 : « curculating » is written instead of circulating

Thank you. This has been corrected (Page 10 line 17).

Line 23 : The author mentioned neutrophils, platelets and monocytes MVs. They also analyzed the endothelial one but do not comment about this population. This should be included.

This was an oversight. This comment has now been included in the Results (Page 10 lines 19 and 36).

Line 50 : figure 3D needs representative images
Representative images are now included.

Page 10

Line 36 : please show the data about cytokine content as supplemental

We cannot include this data as the levels of released cytokines from NMVs were below the detection limit of the cytometric bead array and therefore are blank. We have removed the wording “data not shown” as this may be misleading.

Page 11

Line 27 : Are the authors sure they mean « deduce » ?

We have changed the wording for clarity. The sentence now reads “Therefore, we conclude that NMVs induce NF- κ B activation in endothelial cells via delivery of *miR-155*, which reduces expression of the negative regulator BCL6.”

Line 31 : the quality of the images is not good enough, please split channels to see the BCL6 channel alone (black and with) Line 46 : there is a missing word in the sentence

We have included the BCL6 channel alone in Figure 7b and corrected the sentence (Page 13 line 35).

Discussion Page 12

Lines 23-29 : this paragraph is not clear and I don't understand how this is useful for the discussion of the results. Please clarify.

We agree with the Reviewer that this was not informative. We have removed this section from the discussion.

Figure. 1E. We don't know if this analysis was done at 6w of diet or 20w.

This detail has been added to the figure legend of Figure 1 (Page 21 line 12).

Expanded methods

Multicolor flow cytometry :

After CD144, the authors mentioned « platelet » while this is an endothelial marker.

This has been corrected (Page 2 line 49 of Supplemental Material).

Reviewer #3:

Rodger et al report that Western diet can stimulate the production of circulating micro vesicles in both humans and mice. The microvesicles, particularly those derived from neutrophils (NMV), have an atherogenic effect both in vivo and in vitro. The non- coding small RNA miR155 is one of the cargos further characterized in the neutrophil derived microvesicles. This small non- coding RNA apparently is delivered to endothelial cells, enhances NF-kappaB activation, promotes monocyte adhesion and translocation, possible contributing to plaque formation and inflammation. Overall, the manuscript is well written and the experiments are well designed and comprehensive.

Major concerns:

There is a lack of some important controls. Despite the dialysis to clear the NMV remnants of fMLP, this molecule can have important effects on endothelial cells; for instance, 2,000-fold lower concentration of fMLP than the one used in this work, has been described to induce HUVEC proliferation (Langeggen et al 2001. *Inflammation* (25):83-89). This could play a role in explaining the absence of effect from NMVs prepared from non-stimulated neutrophils.

We thank the Reviewer for highlighting this point. We have tested dialysed supernants for any effect on HCAEC from 3 different donors and could find no significant effect on inflammatory gene expression. This data is included in new Supplemental Figure 10 and detail added to the Methods section (Page 5 line 41).

Are there significantly lower levels of miR155 in the non-stimulated NMVs (I could not find miR155 level comparisons for Unstimulated and stimulated vesicles).

In response to this comment and a comment from Reviewer 2, we have included new data showing the miRNA content of NMVs from unstimulated, fMLP stimulated and acLDL stimulated neutrophils for comparison and showed that *miR-155* levels in NMVs from unstimulated neutrophils are lower compared to those derived from stimulated cells. These data are shown in new Supplemental Figure 13 and the Results section (Page 12, line 44).

How effective are NMVs once mixed with the other MVs, mimicking more effectively what happens in vivo?

We agree with the Reviewer that this mimics more effectively what happens *in vivo* and this is what we investigated in our experiments shown in Figures 8 and 9. NMVs were injected i.v. into *ApoE*^{-/-} mice on Western diet (where increased levels of plasma MVs are circulating (Figure 1h)) and induced an increase in plaque formation.

Is the plasma MV from the individuals exposed to high fat able to inhibit in vitro BCL6 or stimulate RELA in HCAECs?

In this study we have focused on whether NMVs contribute to atherosclerotic plaque formation and the underlying molecular mechanisms. We have not investigated the effects of a mixed population of plasma MVs on HCAEC activation as this would not enable us to focus on characterising the role for NMVs and investigate the enigmatic link between neutrophils and atherosclerosis.

It would be very relevant to quantify if these circulating MVs are present in individuals at risk for atherosclerosis development compared with matched healthy controls.

This is an interesting question. However there are some confounding factors involved in these types of study – the individuals at risk for atherosclerosis will undoubtedly be on treatments to reduce their risk and these treatments may well alter the levels of circulating MVs (as shown by Suades et al., 2013, *Thromb Haemost*, 110, 366-77). At risk subjects often have other co-morbidities such as diabetes, metabolic syndrome and obesity as well as hereditary conditions. These may also have an impact

on MV production (as previously shown by Nomura et al., 2009, *Platelets*, 20, 406-14 for diabetes for example) making interpretation of data challenging. However, since our data clearly show a link between NMVs and atherosclerosis, this may warrant further investigation in a large cohort of well-characterized patients at risk of cardiovascular disease. We have added a section to the discussion regarding this point (Page 15 line 35).

Can the authors at least speculate on what component of the diet, without adding fMLP, would induce NMV production? This is important because unstimulated NMVs do not have proatherogenic effects. Would more physiologic stimuli (Mitochondrial- derived fmLP, oxidized lipoproteins, etc.) have similar effects?

We have added data showing the release, CD18 expression and miR content of MVs in response to modified LDL compared to unstimulated and fMLP stimulated neutrophils (Supplemental Figures 14, 8b and 13 respectively). We have added more discussion about what we speculate to be the components in the diet that may induce MV production (Page 15 line 21).

Minor concerns:

1. A picture of the murine NMV with their size characterization is missing.

This is now included in Figure 1d-f and detail added to the Methods section (Page 5, line 53).

2. Stats in which a t-test is used require normal distribution of the data.

Normal distribution was assumed for the statistical analysis. This detail has now been included in the statistical analysis section (Page 9 line 13).

3. In Fig.2 is difficult to see the bars sizes.

The bars have been increased in size.

4. Fig.3 resolution of fluorescence pictures is not good some of them are pixelated.

We have attempted to improve the resolution. However, some of the images are zoomed in rather than images that were taken at higher magnification.

5. Fig.5 D the green background in the image its interfering with proper visualization of NMVs.

The green background on this image is due to the elastin autofluorescence from smooth muscle cells and is difficult to remove in these types of *in vivo* images.

Fig.6A are these small RNAs are the only ones detected or are they the only ones available for the analysis? Fig6B the difference in miR155 is not particularly high, are the pre-diet MVs able to activate the endothelial cells? Fig5.C and D should those levels of miR155 be in D similar or less than in MV?

We only investigated miRNAs that were not expressed at high levels constitutively in HCAECs, that were known to be increased in activated neutrophils and had been shown to play a role in atherosclerosis or inflammation. This information has been added to the Results section (Page 12 line 37). Pre-diet MVs were able to activate endothelial cells and these were essentially the equivalent to the MVs isolated from healthy volunteers not exposed to HFD used in Figure 4 showing the effect of NMVs on HCAEC gene and protein expression. We have included the basal copy number of *miR-155* in HCAEC as a dotted line on Figure 6f to indicate that pre-diet NMVs are able to increase levels of *miR-155*. Figure 6d shows that in terms of copy number per MV, NMVs have comparatively more *miR-155* than the plasma MVs as a whole. We have added this information into the Results section (Page 12 line 51).

In the supplementary tables, indicate the cell associated with the marker measure and show the averages for each column.

We thank the Reviewer for this useful suggestion. This detail has been included in the tables.

Reviewers' Comments:

Reviewer #1:

Remarks to the Author:

The authors have address my previous concerns performing numerous experiments

Reviewer #2:

Remarks to the Author:

Gomez et al give a clear reply to most of my concerns and questions. This additional work improve the manuscript and highlight the significance of the work for the community. The discussion is greatly improved.

I have few comments left :

Major comments :

Results :

Suppl 11a, page 12, lines 10-11

"Importantly, NMVs released by unstimulated neutrophils were unable to induce a significant alteration in gene expression in HCAECs (Supplemental Figure 11a) suggesting that only NMVs released from stimulated neutrophils are able to induce endothelial cell activation."

This is a wrong statement in regards to the figure presented that I haven't spotted on the first revision round. The figure shows that there is more mRNA in HCAECs treated with fMLP compared to unstimulated MVs not that unstimulated MV do not increase mRNA level in comparison to unstimulated cells.

Is there a normalization step compared to unstimulated cells here that do not clearly appears on the graph? If so, it is important to mentioned it in the legend or on the figure (dotted line for unstim cells for exemple) and if not, the author need to rephrase the sentence or show the right information.

If the authors do not have this information, it would be important to do the corresponding experiments.

Figure suppl13 and 14, page 12, lines 45-50

It is unclear if the mir155 level is expressed as copy number per NMVs or per neutrophils or in a given volume of supernatant. Please clarify.

This raise a question while combining figure suppl13 and 14, if suppl13 is not expressed as a copy number /MV, how the author can state that the NMVs produced upon stimulation contain more mir155? it could be that because the neutrophils release more MVs, the total amount of mir155 detected is increased.

Figure 7

The black and white picture should be inverted to have the signal in black on a white background for a better visibility.

Looking at it for a second time, I am surprise to see a dotty staining while expecting a cytoplasm/nuclear staining. Could the author provide (at least to the reviewers) the negative control (ie. Secondary antibody alone) ? did they performed WB on aortic tissue or qPCR that could confirm this staining?

If not I would rather remove this experiment than showing an uncertain staining (BCL6 is already shown in vitro and is not the main focus here).

Minor comments :

Methods :

Page 5 line 29

The authors refer to figure suppl1 as histology, from the figure suppl 1 it appears unclear if it is histology or cytometry or just hematocrit. Could the authors clarify and refer to the methods required in the methods section if needed ?

Results :

Suppl7, page 10 line 50 and P17 of suppl

The author showed an image from the atheroprone area which is incomplete ("black" area suggesting a fold in the tissue with a z-stack no deep enough to cover the whole sample). As we can expect the labeled MV to cluster in some area based on the image provided in figure 2c, it would be better to provide an image with a complete visibility of the layer. It sounds picky but it could be that by mischance, the unobserved area is where we would see the potential non-specific green dots.

Page 11, line 28

"monocyte transendothelial migration to CCL2" : this sentence seems odd to me, I would rather say "toward CCL2" or "induced by CCL2".

Figure 4d

The flow condition should appear directly on the figure and not only in the legend to facilitate the lecture by the reader.

Figure 5a

It is unclear if the image shown is from a single z-stack, if so, could the authors mentioned it in the legend and provide the stack depth ? this would strengthen the claim on colocalization. If this image is a z-projection, provide a single stack image + the orthonogal view as in 5b.

We thank the reviewer for their meticulous appraisal of our revised manuscript and have addressed their new comments below. We have revised the relevant sections of the manuscript accordingly.

Reviewer #2 (Remarks to the Author):

Gomez et al give a clear reply to most of my concerns and questions. This additional work improve the manuscript and highlight the significance of the work for the community. The discussion is greatly improved.

I have few comments left :

Major comments :

Results :

Suppl 11a, page 12, lines 10-11

“Importantly, NMVs released by unstimulated neutrophils were unable to induce a significant alteration in gene expression in HCAECs (Supplemental Figure 11a) suggesting that only NMVs released from stimulated neutrophils are able to induce endothelial cell activation.” This is a wrong statement in regards to the figure presented that I haven’t spotted on the first revision round. The figure shows that there is more mRNA in HCAECs treated with fMLP compared to unstimulated MVs not that unstimulated MV do not increase mRNA level in comparison to unstimulated cells.

Is there a normalization step compared to unstimulated cells here that do not clearly appears on the graph? If so, it is important to mentioned it in the legend or on the figure (dotted line for unstim cells for exemple) and if not, the author need to rephrase the sentence or show the right information.

If the authors do not have this information, it would be important to do the corresponding experiments.

We thank the reviewer for highlighting this error. We compared levels in endothelial cells incubated with NMVs to levels in HCEAC alone, hence the data is displayed as fold change relative to the cells alone. We have added this information to the figure legend (Supplemental Material Page 21, Lines 8-9) and have altered the text in the results section of the manuscript to more accurately reflect the data (Page 12, Lines 10-14).

Figure suppl13 and 14, page 12, lines 45-50

It is unclear if the mir155 level is expressed as copy number per NMVs or per neutrophils or in a given volume of supernatant. Please clarify.

This raise a question while combining figure suppl13 and 14, if suppl13 is not expressed as a copy number /MV, how the author can state that the NMVs produced upon stimulation contain more mir155? it could be that because the neutrophils release more MVs, the total amount of mir155 detected is increased.

We investigated miR content in the same number (2×10^6) of NMVs isolated from neutrophil exposed to PBS, fMLP and AcLDL. Therefore any change in miRNA levels was not due to differences in the number of

NMVs produced by neutrophils in response to different stimuli. This missing information has now been added to the Figure legend of Supplemental Figure 13 to aid interpretation (Page 23, Line 4).

Figure 7

The black and white picture should be inverted to have the signal in black on a white background for a better visibility. Looking at it for a second time, I am surprised to see a dotted staining while expecting a cytoplasm/nuclear staining. Could the author provide (at least to the reviewers) the negative control (ie. Secondary antibody alone) ? did they performed WB on aortic tissue or qPCR that could confirm this staining? If not I would rather remove this experiment than showing an uncertain staining (BCL6 is already shown in vitro and is not the main focus here).

We have inverted the signal and replaced the original Figure 7. We did perform negative control experiments using secondary antibody alone. Below is an example of the staining we observed. Using the same microscope settings as used to obtain the images in Figure 7 we could not detect any signal in the BCL6 channel (black in greyscale inverted image, red in merged).

Minor comments :

Methods :

Page 5 line 29

The authors refer to figure suppl1 as histology, from the figure suppl 1 it appears unclear if it is histology or cytometry or just hematocrit. Could the authors clarify and refer to the methods required in the methods section if needed ?

We apologise for this omission from the methods. We performed differential blood counts on blood smears stained with hematoxylin and eosin as well as total leukocyte counts using a hemocytometer. This detail has now been added to the methods on Page 5 lines 28-32.

Results : Suppl7, page 10 line 50 and P17 of suppl

The author showed an image from the atheroprone area which is incomplete ("black" area suggesting a fold in the tissue with a z-stack not deep enough to cover the whole sample). As we can expect the labeled MV to cluster in some

area based on the image provided in figure 2c, it would be better to provide an image with a complete visibility of the layer. It sounds picky but it could be that by mischance, the unobserved area is where we would see the potential non-specific green dots.

We have replaced the image in Supplemental Figure 7 to show an atheroprone region of the aorta with all cells stained and a complete monolayer.

Page 11, line 28 “monocyte transendothelial migration to CCL2” : this sentence seems odd to me, I would rather say “toward CCL2” or “induced by CCL2”.

We have replaced “to” with “toward”.

Figure 4d The flow condition should appear directly on the figure and not only in the legend to facilitate the lecture by the reader.

This has now been added to the Figure.

Figure 5a

It is unclear if the image shown is from a single z-stack, if so, could the authors mentioned it in the legend and provide the stack depth ? this would strengthen the claim on colocalization. If this image is a z-projection, provide a single stack image + the orthonogal view as in 5b.

This image was a single z-stack at a depth of 0.8 μm from the base of the cell. This information has now been added to the Figure legend (Page 28, Lines 2-3).

Reviewers' Comments:

Reviewer #2:

Remarks to the Author:

All my remaining concerns have been addressed by the authors.
thank you